# Splines-Based Feature Importance in Kolmogorov-Arnold Networks: A Framework for Supervised Tabular Data Dimensionality Reduction

## Abstract

Feature selection is a key step in many tabular prediction problems, where multiple candidate variables may be redundant, noisy, or weakly informative. We investigate feature selection based on Kolmogorov-Arnold Networks (KANs), which parameterize feature transformations with splines and expose per-feature importance scores in a natural way. From this idea we derive four KAN-based selection criteria (coefficient norms, gradient-based saliency, and knockout scores) and compare them with standard methods such as LASSO, Random Forest feature importance, Mutual Information, and SVM-RFE on a suite of real and synthetic classification and regression datasets. Using average F1 and $R^2$ scores across three feature-retention levels (20%, 40%, 60%), we find that KAN-based selectors are generally competitive with, and sometimes superior to, classical baselines. In classification, KAN criteria often match or exceed existing methods on multi-class tasks by removing redundant features and capturing nonlinear interactions. In regression, KAN-based scores provide robust performance on noisy and heterogeneous datasets, closely tracking strong ensemble predictors; we also observe characteristic failure modes, such as overly aggressive pruning with an $\ell_1$ criterion. Stability and redundancy analyses further show that KAN-based selectors yield reproducible feature subsets across folds while avoiding unnecessary correlation inflation, ensuring reliable and non-redundant variable selection. Overall, our findings demonstrate that KAN-based feature selection provides a powerful and interpretable alternative to traditional methods, capable of uncovering nonlinear and multivariate feature relevance beyond sparsity or impurity-based measures.

## 1 Introduction

Feature selection plays a central role in many machine learning pipelines, where models must cope with redundant, noisy, or weakly informative input variables. By identifying and retaining only the most relevant features, it can improve both interpretability and computational efficiency. Classical approaches to feature selection are often grouped into three categories: *filter* methods, which rank features using data-driven scores independent of the predictor; *wrapper* methods, which evaluate subsets of features through repeated model training; and *embedded* methods, which perform selection as part of the model fitting procedure itself.

Filter methods include statistical criteria such as Pearson correlation and mutual information (MI) (Peng et al., 2005), which are fast and model-agnostic but ignore feature interactions and often select redundant features. Embedded methods integrate selection into the learning algorithm itself, such as Least Absolute Shrinkage and Selection Operator(LASSO) (Tibshirani, 1996a), spline-penalized regression (Marx & Eilers, 1996), tree-based models like Random Forests (Breiman, 2001), and gradient-boosted methods such as XGBoost (Chen & Guestrin, 2016), all of which provide model-aware importance scores but may sacrifice interpretability in complex non-linear settings. Wrapper methods, such as Recursive Feature Elimination (RFE) for Support Vector Machines (SVMs) (Guyon et al., 2002), directly search for subsets of features by iteratively training and evaluating

models; these often yield strong predictive performance but are computationally expensive and prone to overfitting. Recent heuristic ensemble strategies such as the Minimum Union (Min) method (Radovic et al., 2017) have also been proposed to combine outputs of multiple selectors, improving stability but adding complexity. Finally, unsupervised projection techniques like Principal Component Analysis (PCA) (Jolliffe, 2002) reduce dimensionality by constructing orthogonal linear combinations of the original variables, which improves computational efficiency but discards direct interpretability and neglects target information. Overall, while traditional methods offer a rich toolbox for feature selection, they involve trade-offs among computational cost, predictive performance, stability, and interpretability. Filter methods are fast but myopic, wrappers are accurate but expensive, and embedded methods are model-aware but often opaque. These limitations motivate the exploration of new architectures, such as Kolmogorov-Arnold Networks (KANs), which offer both model-awareness and direct interpretability by parameterizing each feature transformation as a trainable spline.

Kolmogorov-Arnold Networks (KANs) are a recently proposed neural architecture (Liu et al., 2024b) inspired by the Kolmogorov-Arnold representation theorem, which states that any continuous multivariate function can be expressed as a finite superposition of univariate functions and addition. KANs operationalize this idea by replacing traditional scalar weights with trainable spline functions: each input feature is passed through one-dimensional, edge-specific splines, aggregated at hidden nodes, and then transformed by learnable outer splines. This "weights-as-functions" paradigm yields compact yet expressive models and, crucially for feature selection, a structured parameterization in which all parameters associated with a given feature form an explicit block. Intuitively, if the learned splines attached to feature $j$ are nearly flat, then the network's output varies little with respect to that coordinate, whereas large or highly curved splines signal a strong, nonlinear dependence. KANs therefore provide a natural basis for defining feature-importance scores directly from a trained model.

Despite this appealing structure, existing work on KANs for feature selection remains sparse and does not fully exploit these properties. Zheng et al. (2025) use KAN as a surrogate fitness evaluator within a Whale Optimization Algorithm for high-dimensional medical data, inheriting the computational cost and sensitivity of meta-heuristic search and treating KAN largely as a black-box scorer. Other preliminary studies rely on manual spline inspection to prune features (Wang et al., 2025) or use KAN as a front-end encoder for IMU time series(Liu et al., 2024a), without turning its spline parameters and gradients into systematic importance measures.

In this work, we close this gap by introducing a family of KAN-based feature-importance criteria that explicitly leverage the weights-as-functions view. We derive coefficient-based norms of spline parameters (*KAN-L1*, *KAN-L2*) to capture the global magnitude of each feature's learned univariate transformation, a sensitivity integral based on input gradients (*KAN-SI*) to quantify local influence, and a knock-out score (*KAN-KO*) that measures the increase in loss when a feature's spline block is ablated. We then systematically evaluate these KAN-based selectors as supervised feature-selection methods on tabular classification and regression benchmarks, comparing them with representative filter, wrapper, and embedded baselines. Our analysis covers predictive performance (macro-$F_1$ and $R^2$), robustness to redundancy (average pairwise correlation among selected features), stability (Jaccard similarity of selected sets across the five cross-validation folds), and interpretability (plots of KAN layer responses and class logits for top-ranked features) [1]. Empirically, *KAN-L2*, *KAN-SI*, and *KAN-KO* are competitive with or superior to classical baselines on structured and synthetic datasets, while remaining robust on noisy real-world tasks; *KAN-L1* can be highly effective in some classification settings but tends to over-prune in regression. Overall, our results indicate that KAN-based selectors provide a practical and interpretable alternative to traditional methods.

The paper is organized as follows. Section 2 surveys the related work on feature selection methods. Section 3 provides background and introduces Kolmogorov-Arnold Networks (KANs). Section 4 presents the methodology: coefficient-based feature importance (KAN-L1, KAN-L2), KAN-Knockout (KAN-KO), and KAN-Sensitivity Integral (KAN-SI), and it specifies the assessment pipeline from feature selection to prediction (predictors, evaluation metrics, and leakage-safe cross-

---

[1] See appendix 6

validation) together with the baseline methods. Section 5 describes the dataset, experimental results and analysis along with a discussion of different results.

## 2  RELATED WORK

In supervised learning with tabular data, feature selection is a critical step for improving predictive performance, enhancing interpretability, and reducing computational cost. High-dimensional datasets, common in domains such as finance, bioinformatics, and environmental monitoring often contain irrelevant or redundant variables that can degrade model accuracy and increase overfitting risk. Over time, a variety of supervised feature selection techniques have been developed.

One widely used embedded method is the Least Absolute Shrinkage and Selection Operator (LASSO) (Tibshirani, 1996b), which adds an $\ell_1$ penalty to shrink small coefficients to zero, yielding sparse and interpretable linear models at modest computational cost. However, in the presence of strong collinearity it typically keeps only one predictor from a correlated group, potentially discarding relevant variables (Zou & Hastie, 2005). Generalized Additive Models (GAMs) (Wood & Augustin, 2002) extend linear models with penalized splines to capture nonlinear effects while retaining some interpretability, but fitting and selecting splines becomes computationally demanding as dimensionality grows. Tree-based ensembles such as Random Forests (Breiman, 2001) provide impurity-based importance scores and naturally capture interactions, yet their importance can be biased toward high-variance or high-cardinality features and need not transfer well to other model classes (Strobl et al., 2007). Wrapper methods like Recursive Feature Elimination (RFE) (Guyon et al., 2002) iteratively retrain a model while pruning low-importance features, often achieving strong accuracy but at significant computational cost. Finally, information-theoretic filter methods such as Mutual Information (MI) ranking (Peng et al., 2005) can detect nonlinear dependencies without repeated model training, but because they score features individually, they do not account for redundancy and tend to select correlated variables with overlapping information.

Extreme Gradient Boosting (XGBoost) (Chen & Guestrin, 2016) is a widely used tree-ensemble method noted for its efficiency and built-in feature importance metrics. During training, XGBoost performs embedded feature selection by evaluating candidate splits and produces importance scores (e.g., gain or split counts) that can be used to rank and filter variables; this has been exploited in numerous applications, such as reducing a 42-dimensional intrusion-detection dataset to 19 informative features with improved accuracy (Kasongo & Sun, 2020). SHAP (SHapley Additive exPlanations) (Lundberg & Lee, 2017) extends this idea by providing theoretically grounded, instance-level attributions whose magnitudes can be aggregated for feature ranking, often yielding more interpretable importance profiles and supporting domain-facing explanations (e.g., in medical risk models). Empirical studies, however, highlight trade-offs: Wang et al. (2024) report that simple XGBoost importance-based selection can outperform SHAP-based selection in both AUPRC and computational efficiency on credit fraud data, reflecting the additional overhead of SHAP computation.

Minimum Union (Min) Method (Seijo-Pardo et al., 2019) aggregates multiple feature ranking results to improve stability in high-dimensional tabular data. The Min method computes each feature's best (lowest) rank across an ensemble of selectors, yielding a combined ranking that favors features identified as important by any selector (Seijo-Pardo et al., 2016). However, the evaluation of Min has largely been confined to such high-dimensional biological data, and it has not been extensively benchmarked on diverse datasets or against a wide range of modern feature selection methods. Thus, while Min can improve selection stability and performance in some scenarios, its general efficacy across other domains and in combination with different learning algorithms remains an open question in the literature.

Beyond the original KAN formulation(Liu et al., 2024b; Akazan et al., 2025), recent variants such as GKAN lifts KANs to graph-structured domains, learning node/graph functions via Kolmogorov-Arnold compositions on topology-aware inputs (Kiamari et al., 2024). In biomedicine, interpretable Graph KANs fuse multi-omics with graph priors to achieve multi-cancer classification and biomarker discovery, emphasizing model transparency through graph-aware architectures (Alharbi et al., 2025). Complementarily, KAN-guided Whale Optimization uses KAN outputs to steer a metaheuristic for feature selection on medical datasets, reporting empirical gains but relying on heuristic search rather than first-principles criteria (Zheng et al., 2025). Our contribution is orthog-

onal: we introduce a generic KAN feature-importance framework (KAN-SI,KO, L1 and L2) that derives importance from explicit chain-rule sensitivities and spline energy, treats categoricals in a reference-invariant manner, and separates relevance from redundancy through a derivative-Gram audit enabling diversity-aware selection.

## 3 BACKGROUND

This part provides a detailed analysis of the vanilla KAN, which we used for our study.

### 3.1 KOLMOGOROV-ARNOLD NETWORKS

KANs (Liu et al., 2024b; Akazan et al., 2025) are inspired by Kolmogorov's superposition theorem, which states that any multivariate continuous function $f(x_1, \ldots, x_n)$ can be represented as a finite composition of continuous univariate functions. KAN models $f : \mathbb{R}^n \to \mathbb{R}$ as:

$$f(\mathbf{x}) = \sum_{j=1}^{2n+1} \phi_j \left( \sum_{i=1}^{n} \psi_{ij}(x_i) \right), \tag{1}$$

where $\psi_{i,j} : [0,1] \to \mathbb{R}$ are the univariate functions (B-splines in this case) $\phi_j : \mathbb{R} \to \mathbb{R}$ are continuous functions.

Each input feature $x_i$ is passed through a spline basis transformation:

$$\psi_{ij}(x_i) = \sum_{k=1}^{K} w_{ijk} B_k(x_i), \tag{2}$$

where $B_k$ is a fixed spline basis (e.g., B-splines or sinusoidal), and $w_{ijk}$ are learnable coefficients. These coefficients are learned via backpropagation.

**KAN layer (matrix form).** Let $x \in \mathbb{R}^d$ be an input vector and $m$ the number of outputs. A KAN layer produces

$$y = W_{\text{base}} \, \phi(x) + \sum_{j=1}^{d} W_{\text{spline}}^{(j)} \, b_j(x_j) \in \mathbb{R}^m, \tag{3}$$

where: $W_{\text{base}} \in \mathbb{R}^{m \times d}$ is the base (linear) weight matrix; $\phi : \mathbb{R}^d \to \mathbb{R}^d$ is applied componentwise (e.g. SiLU), so $(\phi(x))_j = \phi(x_j)$; $b_j(x_j) \in \mathbb{R}^K$ is the $K$-dimensional B-spline basis vector for feature $j$; $W_{\text{spline}}^{(j)} \in \mathbb{R}^{m \times K}$ are the spline weights attached to feature $j$.

Stacking all spline bases into a single vector

$$b(x) = \begin{bmatrix} b_1(x_1)^\top b_2(x_2)^\top \cdots b_d(x_d)^\top \end{bmatrix}^\top \in \mathbb{R}^{d \times K}, \tag{4}$$

and concatenating weights

$$W_{\text{spline}} = \begin{bmatrix} W_{\text{spline}}^{(1)} W_{\text{spline}}^{(2)} \cdots W_{\text{spline}}^{(d)} \end{bmatrix} \in \mathbb{R}^{m \times d \times K}, \tag{5}$$

the layer is simply

$$y = W_{\text{base}} \, \phi(x) + W_{\text{spline}} \, b(x). \tag{6}$$

**Batch form.** For $X \in \mathbb{R}^{n \times d}$ (rows are samples), define $\Phi(X) \in \mathbb{R}^{n \times d}$ by $(\Phi(X))_{ij} = \phi(X_{ij})$, and $B(X) \in \mathbb{R}^{n \times d \times K}$ by concatenating $b_j(X_{:j})$ columnwise. Then the output matrix $Y \in \mathbb{R}^{n \times m}$ is

$$Y = \Phi(X) \, W_{\text{base}}^\top + B(X) \, W_{\text{spline}}^\top. \tag{7}$$

## 4 METHODOLOGY

In a KAN, each input coordinate is passed through a small set of one-dimensional spline functions and then combined linearly, so all parameters associated with a given feature form an explicit, low-dimensional block. This structure naturally supports several ways of quantifying how much each input dimension contributes to the learned mapping. We therefore define four KAN-based feature-importance criteria derived from a trained model. This section discusses our four KAN-based selectors. [2]

### 4.1 COEFFICIENT-BASED FEATURE IMPORTANCE (KAN-L1 AND KAN-L2)

In a KAN layer, each input feature $x_i$ is expanded into a set of $K$ B-spline basis functions,

$$z_i = \sum_{k=1}^{K} w_{ik}\, B_k(x_i), \tag{8}$$

where $w_{ik}$ are the learned spline coefficients for feature $x_i$. These coefficients capture how strongly the model relies on different local regions of $x_i$'s domain.

To quantify feature importance for $x_i$, we aggregate each coefficient vector $\mathbf{w}_i = (w_{i1}, \ldots, w_{iK})$ using its $\ell_1$ and $\ell_2$ norms:

$$I_{L_1}(x_i) = \|\mathbf{w}_i\|_1 = \sum_{k=1}^{K} |w_{ik}|, \qquad I_{L_2}(x_i) = \|\mathbf{w}_i\|_2 = \left(\sum_{k=1}^{K} w_{ik}^2\right)^{1/2}. \tag{9}$$

To enable comparability across features, we normalize these scores to create the final importance scores:

$$\tilde{I}_{L_1}(x_i) = \frac{I_{L_1}(x_i)}{\sum_{j=1}^{d} I_{L_1}(x_j)}, \qquad \tilde{I}_{L_2}(x_i) = \frac{I_{L_2}(x_i)}{\sum_{j=1}^{d} I_{L_2}(x_j)}. \tag{10}$$

These normalized quantities provide a direct and interpretable measure of how much each input contributes, on average, through its spline expansion.

### 4.2 KAN-KNOCKOUT (KAN-KO)

Let the trained KAN layer (see Subsection 3.1) be parameterized by $W := (W_{\text{base}}, W_{\text{spline}})$. For a given feature index $j \in \{1, \ldots, d\}$, define the knockout operator $\mathcal{K}_j$ acting on the first KAN layer by

$$\mathcal{K}_j(W) := \big(\widetilde{W}_{\text{base}}, \widetilde{W}_{\text{spline}}\big), \quad \text{where} \quad \widetilde{W}_{\text{base}}(:,j) = 0, \widetilde{W}_{\text{spline}}^{(j)} = 0. \tag{11}$$

Equivalently, at the level of the layer output

$$y^{(-j)}(x) = y(x) - W_{\text{base}}(:,j)\, \phi(x_j) - W_{\text{spline}}^{(j)}\, b_j(x_j). \tag{12}$$

To quantify the contribution of each input feature, we evaluate how much the task loss increases when that feature is removed from the model. Let $\ell(\hat{y}, y)$ denote the task loss and $P$ the data distribution over $(x, y)$. We write $f_W$ for the full KAN model, where $W$ collects the base and spline weights (see Subsection 3.1). The expected or population risk of the model is

$$L(W) = \mathbb{E}_{(x,y)\sim P}\big[\,\ell(f_W(x), y)\,\big]. \tag{13}$$

For a given feature index $j$, we define a knock-out operator $\mathcal{K}_j(W)$ that sets to zero the $j$th column of $W_{\text{base}}$ and the $j$th spline block of $W_{\text{spline}}$. The corresponding risk when feature $j$ is removed is

$$L_j(W) = \mathbb{E}_{(x,y)\sim\mathcal{D}}\big[\,\ell(f_{\mathcal{K}_j(W)}(x), y)\,\big]. \tag{14}$$

The *KAN-KO importance score* for feature $j$ is the nonnegative increase in risk,

$$\Delta_j = \max\big\{0, L_j(W) - L(W)\big\}. \tag{15}$$

To make scores comparable across features, we normalize them as

$$I_j = \frac{\Delta_j}{\sum_{k=1}^{d} \Delta_k + \delta}, \tag{16}$$

where $\delta > 0$ is a small constant to avoid division by zero.

---

[2] See the appendix A for more details

### 4.3 KAN- SENSITIVITY INTEGRAL (KAN-SI) FEATURE IMPORTANCE

For an input feature $x_i$, the instantaneous (local) sensitivity of $f$ is the magnitude of its partial derivative,

$$S_i(\mathbf{x}) = \left| \frac{\partial f(\mathbf{x})}{\partial x_i} \right|. \tag{17}$$

The KAN-SI importance is the (data) expectation of $S_i$:

$$I_i = \mathbb{E}_{\mathbf{x}}[S_i(\mathbf{x}), \qquad \widehat{I}_i = \frac{1}{N} \sum_{n=1}^{N} \left| \frac{\partial f(\mathbf{x^n})}{\partial x_i} \right| \tag{18}$$

where $\widehat{I}_i$ is the empirical estimator over a held-out set $\{\mathbf{x}^n\}_{n-1}^N$ (validation or out-of-fold) to avoid optimistic bias.

From equation 1, let the inner pre-activations be

$$t_j(\mathbf{x}) = \sum_{i=1}^{n} \psi_{ij}(x_i).$$

Then by the chain rule,

$$\frac{\partial f(\mathbf{x})}{\partial x_i} = \sum_{j=1}^{2n+1} \phi_j'(t_j(\mathbf{x})) \frac{\partial \psi_{ij}(x_i)}{\partial x_i}. \tag{19}$$

Because $\psi_{ij}$ is a spline expansion,

$$\frac{\partial \psi_{ij}(x_i)}{\partial x_i} = \sum_{k=1}^{K} w_{ijk} B_k'(x_i), \tag{20}$$

and for B-splines of degree $p$, only $p+1$ basis derivatives $B_k'(x_i)$ are non-zero at a given $x_i$ (local support), which makes equation 19 efficient to evaluate.

Since the sensitivities depend on the measurement of $x_i$, we report a normalized score.

$$\tilde{I}_i = s_i \widehat{I}_i, \qquad s_i \in \{ \mathrm{Std}(x_i), \mathrm{IQR}(x_i) \} \tag{21}$$

where $\mathrm{Std}$(standart deviation) or $\mathrm{IQR}$(interquartile range) is computed on the same split used for equation 18. For one - hot encoded categoricals, compute $\tilde{I}$ per dummy and *sum* back to the original category. KAN-SI quantifies the importance of feature $x_i$ as the expected on-data magnitude of the model's directional derivative given by equation 18 and report the scale-invariant score using equation 21 to neutralize unit effects.

### 4.4 ASSESSMENT PROCEDURE (FROM FEATURE SELECTION TO PREDICTION)

Let $D = \{(x_i, y_i)\}_{i=1}^n$, where $x_i \in \mathbb{R}^d$ and

$$y_i \in \begin{cases} \{1, \ldots, C\}, & \text{classification,} \\ \mathbb{R}, & \text{regression.} \end{cases} \tag{22}$$

Let $\mathcal{S} = \{s_1, \ldots, s_L\}$ denote feature selectors (e.g., KAN-L1/L2, KAN-SI/KO, MI, LASSO, etc.). Each $s \in \mathcal{S}$ fitted on training data returns a nonnegative importance vector $a_s \in \mathbb{R}_+^d$, normalized so $\sum_{j=1}^d a_{s,j} = 1$. For a retention ratio $k \in \{20\%, 40\%, 60\%\}$, set

$$n_k = \max\left\{ 1, \left\lceil \frac{k \cdot d}{100} \right\rceil \right\}, \text{number equivalent of k\% of the features} \tag{23}$$

Let $J_{s,k} \subset \{1, \ldots, d\}$ be the indices of the $n_k$ largest entries of $a_s$ (ties arbitrary), and define the *projection* $\Pi_J(x) \in \mathbb{R}^{|J|}$ that restrict $x$ to the matrix defined by the coordinates $J$.

### 4.4.1 PREDICTORS AND EVALUATION METRICS.

For each task $t \in \{\text{classification}, \text{regression}\}$, we consider a family of predictors $\mathcal{P}_t = \{P_{t,1}, \ldots, P_{t,M}\}$, including Logistic Regression / Ridge, Random Forests, Gradient Boosted Trees (GBT), and XGBoost. Predictive performance is quantified by

$$S_t(y, \hat{y}) = \begin{cases} \text{Macro-}F_1(y, \hat{y}), & t = \text{classification}, \\ R^2(y, \hat{y}), & t = \text{regression}, \end{cases} \tag{24}$$

where the per-class $F_1$ score is defined one-vs-rest and averaged across $C$ classes:

$$F1_c = \frac{2TP_c}{2TP_c + FP_c + FN_c}, \qquad \text{Macro-}F_1 = \frac{1}{C}\sum_{c=1}^{C} F1_c. \tag{25}$$

### 4.4.2 LEAKAGE-SAFE CROSS-VALIDATION.

We evaluate each selector-predictor pair using leakage-safe $F$-fold cross-validation. For each fold $f$, we fit preprocessing and the feature selector $s$ *only* on the training split $T_f$, obtain the selected feature set $J_{s,k}^{(f)}$, train the predictor on the projected training data, and then compute the score $S_{s,k,t,m,f}$ (macro-$F_1$ or $R^2$) on the projected validation split $V_f$. The overall performance $\text{Score}(s, k, t, m)$ is the average of these fold-level scores. This procedure ensures that both feature selection and model training see only training data in each fold, preventing information leakage from the validation sets and yielding an unbiased estimate of generalization performance. [3]

**Comparison with Baseline Methods**   We compare our spline-based feature selectors against Mutual Information (MI) ranking (Peng et al., 2005), Random Forest (Breiman, 2001), LASSO-based feature selection (Tibshirani, 1996b), and Recursive Feature Elimination (RFE) for Support Vector Machine (SVM) (SVM-RFE)(Guyon et al., 2002).

## 5   EXPERIMENTS

Reproducibility details and data sets information can be found at Appendix B.1 and Appendix B.2.

The comparative analysis across datasets reveals that averaging F1 and $R^2$ scores across retention levels $k$ reveals consistent yet dataset-specific interactions between selectors, predictors, and the underlying data structure.

**Classification datasets**   On the *Breast Cancer* dataset (Figure 1), Gradient Boosted Trees perform best with *KAN-L1*, LASSO, and RF importance, often matching the full-feature baseline. In contrast, *KAN-KO* and Mutual Information yield poor subsets, while Logistic Regression and tree ensembles (RF, XGBoost) benefit most from sparsity- or impurity-based selectors. This indicates that selectors capturing either linear sparsity or tree-based splits are most effective, whereas knock-out and sensitivity KAN variants are less suited. In the *Digits* dataset (Figure 1), the pattern shifts: Gradient Boosted Trees and XGBoost achieve higher F1 scores with LASSO, SVM-RFE, and several KAN variants (*KAN-KO*, *KAN-SI*, *KAN-L1*), sometimes surpassing the full-feature baseline. Here, dimensionality reduction removes redundancy and sharpens predictive power. By contrast, *KAN-L2* and Mutual Information underperform, reflecting their limits in capturing complex feature interactions. For the synthetic *make_classification* dataset (Figure 1), all KAN-based selectors (*KAN-L1/L2*, *KAN-SI*, *KAN-KO*) consistently outperform classical baselines, confirming their advantage in structured, high-signal settings where ground-truth feature relevance extends beyond sparsity or univariate dependence. Finally, on the *Wine* dataset (Figure 1), the advantage shifts back toward classical selectors: RF importance, LASSO, Mutual Information, and SVM-RFE provide the strongest subsets, while most KAN-based selectors lag behind. Only *KAN-SI* remains competitive, suggesting that when features are few and relationships are well-structured, simpler selectors aligned with linear sparsity or impurity-based measures are more effective.

---

[3]See the mathematical formulation of this process in appendix B.3

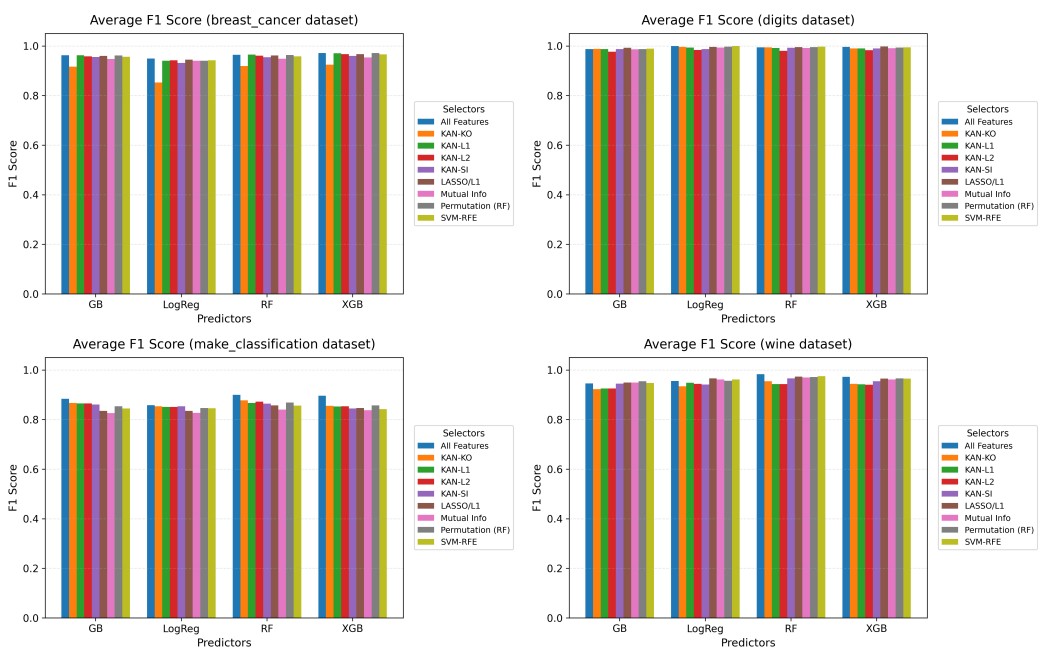

Figure 1: Average $F1$ Score ( averaged across 20/40/60% retained features) per Selectors for Each Classifiers

**Regression datasets** On the *California dataset (nonlinear, heterogeneous), Figure. 2*, *KAN-L2* and *KAN-SI* preserve tree-ensemble performance: with Random Forest and XGBoost they stay close to the orediction using all Features (minor drops ∼0.01-0.03), while Gradient Boosted Tree is slightly more sensitive. In contrast, *KAN-L1* is overly aggressive: it depresses Ridge markedly

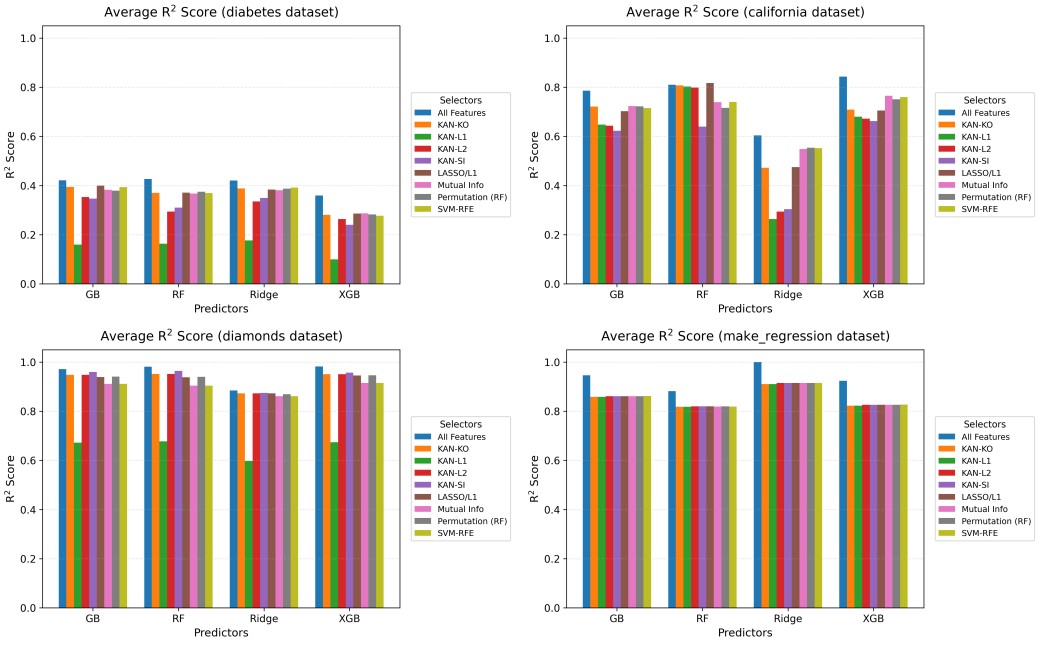

Figure 2: Relative Average $R^2$ Score (averaged across 20/40/60% retained features) per Selectors for Each Regressors

(large $R^2$ gap vs. All Features) and also trims XGBoost and Gradient Boosted Tree more than other KAN variants, indicating loss of weak but complementary signals. *KAN-KO* feature selector is generally competitive, trailing All Features by a small margin and often matching the classical Mitual Info, Random Forest and SVM-RFE subsets. For *Diabetes dataset (small-$n$, noisy), Figure. 2)*, *KAN-L1* again under-selects (lowest bars for Gradient Boosted Tree, Random Forest, XGBoost), whereas *KAN-L2,KAN-SI* yield the most stable KAN performance across predictors and typically track LASSO and Mutual Info. *KAN-KO* behaves as a conservative KAN subset, usually second-best among KANs. In the *Diamond dataset (strong signal, structured, mixed types), Figure. 2,* All models are high-$R^2$; here *KAN-KO,KAN-L2* and *KAN-SI* are nearly indistinguishable from All Features for Random Forest, XGBoost and Gradient Boosted Tree and even outperform classical features selectors (Mitual Info, Random Forest and SVM-RFE subsets, LASSO and Random Forest). *KAN-L1* is the only KAN variant that noticeably drops on some predictors, consistent with over-pruning informative but correlated attributes. For the Synthetic *make regression dataset (well-conditioned, Figure. 2,* All KAN variants cluster tightly with baselines across predictors (tiny spreads): when the signal is clean and features are already informative, KAN selection neither helps nor hurts much; *KAN-L2/KAN-SI* remain the safest choices.

Taken together, these findings suggest that KAN-based selectors are particularly attractive when feature interactions and nonlinearity play a central role, or when one wishes to couple selection and interpretability (see Appendix C). Their main failure mode is over-pruning correlated yet informative features under an $\ell_1$-style criterion (*KAN-L1*), in regression. More broadly, the results reinforce that dimensionality reduction is not uniformly harmful: when a selector is aligned with the structure of the data and the bias of the predictor, removing redundant or spurious variables can improve both accuracy and interpretability.

## 6 CONCLUSION

This work presented, to our knowledge, the first systematic study of Kolmogorov-Arnold Network (KAN) based feature selection on tabular classification and regression benchmarks. We defined four KAN-derived criteria (*KAN-L1*, *KAN-L2*, *KAN-SI*, *KAN-KO*) and compared them against widely used baselines, including LASSO, Random Forest feature importance, mutual information, and SVM-RFE, across multiple datasets and feature-retention levels. Empirically, *KAN-L2*, *KAN-SI*, and *KAN-KO* provide competitive and often superior performance in structured or strong-signal settings, while remaining robust on real-world tasks; *KAN-L1* can be effective in classification but tends to over-prune in noisy or correlated regression problems. Classical selectors such as LASSO and Random Forest remain strong choices, particularly when they align with the assumptions of the downstream model. Furthermore, our stability and redundancy analyses show that KAN-based criteria produce reproducible feature subsets across folds and do not inflate correlation among selected features indicating that they offer both reliable and non-redundant selection even without enforcing sparsity constraints. Beyond aggregate scores, our spline-based case studies show that KANs can yield smooth, one-dimensional response functions that link feature values to model behaviour in a transparent way, offering an interpretable view of nonlinear and multivariate relevance that is difficult to obtain from purely sparsity- or impurity-based methods. Overall, our findings indicate that KAN-based feature selection is a practical, interpretable alternative to traditional approaches, and they provide guidance on when specific KAN criteria are most beneficial (e.g., *KAN-L2/KAN-SI* for noisy regression, *KAN-SI/KAN-KO* for interaction-heavy classification). We see this as a step toward feature-selection methods that jointly offer strong predictive performance, robustness to irrelevant variables, and clear mechanistic insight into how individual predictors influence model outputs. At the same time, training KANs is noticeably slower than fitting sparse linear models or tree ensembles, which limits their use in large-scale screening or AutoML settings. An important direction for future work is therefore to develop faster, yet still interpretable, KAN architectures and training schemes (e.g., lightweight KAN layers, pruning or distillation of spline components, or hybrid models that reuse KAN-based scores to guide simpler selectors).

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

APPENDIX

# Part I

# Table of Contents

# A  MATHEMATICAL MOTIVATION OF KAN-BASED FEATURE SCORES

This section gives a brief functional motivation for the four KAN-based feature-importance criteria used in the main paper.

## A.1  FUNCTIONAL VIEW OF THE FIRST KAN LAYER

For clarity, consider a single-output KAN with one hidden layer. The first layer applies, for each unit $r = 1, \ldots, R$ and feature $j = 1, \ldots, d$, a univariate spline

$$g_{j,r}(x_j) = \sum_{k=1}^{K} \theta_{j,r,k}\, B_k(x_j), \tag{26}$$

where $\{B_k\}_{k=1}^{K}$ are fixed B-spline basis functions and $\theta_{j,r,k}$ are learned coefficients. Collecting all coefficients attached to feature $j$ in the first layer gives a parameter block

$$\theta_j = (\theta_{j,r,k})_{r=1,\ldots,R;\, k=1,\ldots,K}. \tag{27}$$

The network output can be written schematically as

$$f(x) = \sum_{r=1}^{R} w_r\, \sigma\Big( \sum_{j=1}^{d} g_{j,r}(x_j) \Big), \tag{28}$$

so that feature $j$ influences $f$ only through its spline family $\{g_{j,r}\}_r$ controlled by $\theta_j$. The scores below quantify the importance of feature $j$ by measuring, in different ways, how much the learned function $f$ depends on this block.

## A.2  COEFFICIENT NORMS: *KAN-L1* AND *KAN-L2*

For fixed $j$ and $r$, define the spline $g_{j,r}(x_j) = \sum_k \theta_{j,r,k} B_k(x_j)$ and consider its $L^2(P_X)$ norm with respect to the data distribution $P_X$:

$$\|g_{j,r}\|_{L^2(P_X)}^2 = \mathbb{E}_X\Big[g_{j,r}(X_j)^2\Big] = \theta_{j,r}^\top G_j\, \theta_{j,r}, \tag{29}$$

where $G_j = \mathbb{E}_X[B(X_j)B(X_j)^\top]$ is the Gram matrix of the B-spline basis for feature $j$. For standardized inputs and regular knots, $G_j$ is well-conditioned, so there exist constants $0 < \lambda_{\min} \le \lambda_{\max} < \infty$ such that

$$\lambda_{\min} \|\theta_{j,r}\|_2^2 \le \|g_{j,r}\|_{L^2(P_X)}^2 \le \lambda_{\max} \|\theta_{j,r}\|_2^2. \tag{30}$$

Thus, up to fixed constants, the $\ell_2$ norm of the spline coefficients is proportional to the $L^2$ "energy" of the learned univariate transformation of feature $j$. Summing over hidden units $r$ yields that $\|\theta_j\|_2$ is a proxy for the total contribution of feature $j$ to the first-layer representation. This motivates the *KAN-L2* score as

$$\mathrm{score}_{\mathrm{L2}}(j) = \|\theta_j\|_2. \tag{31}$$

The *KAN-L1* score $\mathrm{score}_{\mathrm{L1}}(j) = \|\theta_j\|_1$ emphasizes sparsity over basis functions: features whose effect is concentrated on a small number of knots receive higher values, encouraging compact, interpretable univariate responses.

## A.3  SENSITIVITY INTEGRAL: *KAN-SI*

Let $f : \mathbb{R}^d \to \mathbb{R}$ be the trained KAN predictor and assume $f$ is differentiable almost everywhere. For a small perturbation $\delta$ along coordinate $j$ we have the first-order approximation

$$f(x + \delta e_j) - f(x) \approx \delta\, \frac{\partial f}{\partial x_j}(x), \tag{32}$$

where $e_j$ is the $j$-th canonical basis vector. Taking expectations over the data distribution and random perturbations with $\mathbb{E}[|\delta|] = c$ gives

$$\mathbb{E}\big[|f(X + \delta e_j) - f(X)|\big] \approx c\, \mathbb{E}_X\Big[\big|\tfrac{\partial f}{\partial x_j}(X)\big|\Big]. \tag{33}$$

The quantity

$$S_j^{\text{SI}} = \mathbb{E}_X\left[\left|\tfrac{\partial f}{\partial x_j}(X)\right|\right] \tag{34}$$

therefore measures the expected local sensitivity of the prediction to perturbations in feature $j$. Our *KAN-SI* score is the empirical estimate of $S_j^{\text{SI}}$ computed on a validation set, using the gradient of the trained KAN. Features with large $S_j^{\text{SI}}$ are those for which small changes in $x_j$ typically induce large changes in the output, making *KAN-SI* a derivative-based global importance measure.

### A.4 KNOCK-OUT LOSS: *KAN-KO*

Let $\ell(f(X), Y)$ denote the training loss (cross-entropy for classification or squared error for regression) and $R(f) = \mathbb{E}[\ell(f(X), Y)]$ its risk. Given a trained KAN $f$, we define $f^{(-j)}$ as the same network but with all spline parameters associated with feature $j$ in the first layer set to zero, so that feature $j$ no longer contributes to the first-layer representation. The *knock-out* importance of feature $j$ is then

$$\Delta_j = R(f^{(-j)}) - R(f) \geq 0, \tag{35}$$

the increase in risk incurred when the contribution of feature $j$ is ablated. In practice, we estimate $\Delta_j$ by the empirical risk difference on a held-out set, using the same trained parameters and only modifying the first-layer spline block of feature $j$. When $f$ approximately minimizes $R$, larger $\Delta_j$ indicate that the model relies heavily on feature $j$ for accurate predictions. This motivates the *KAN-KO* score as a leave-one-feature-out, loss-based measure of relevance.

### A.5 SUMMARY

In summary, the four KAN-based criteria exploit the spline-based parameterization of the first layer to approximate complementary notions of feature relevance: *KAN-L2/KAN-L1* measure the "energy" and sparsity of the learned univariate transformations, *KAN-SI* captures gradient-based global sensitivity, and *KAN-KO* quantifies the increase in risk when a feature's contribution is removed. All are computed directly from a trained KAN, without external wrapper search or additional surrogate models.

## B REPRODUCIBILITY

This section provides necessary reproducibility details

### B.1 REPRODUCIBILITY DETAILS

We implement all models in PyTorch using a Kolmogorov-Arnold Network (KAN) whose architecture is defined by the experiment-specific list *layers_hidden*=$[n_{input}, 2n + 1, n_{output}]$. Across all experiments, the KAN modules rely on cubic spline bases over a grid of five knots on the interval $[-1, 1]$, with a small grid offset of $0.02$ to avoid boundary artefacts. The base branch uses the *SiLU* nonlinearity, while the spline and base components are initialized with unit scaling and a modest amount of injected noise (scale $0.1$) to encourage exploration during early training. Models are trained for 100 epochs with a mini-batch size of 64 using the Adam optimizer with PyTorch's default hyperparameters. All runs are executed on Kaggle's GPU environment with an NVIDIA Tesla P100, using fixed random seeds for Python, NumPy, and PyTorch and identical train-test splits across methods to ensure reproducibility.

### B.2 DATA SETS

To evaluate the effectiveness of the Kolmogorov-Arnold Networks dimensinality reduction methods, experiments were conducted on a diverse collection of benchmark datasets covering both classification and regression tasks. These datasets includes well-established repositories from scikit-learn as well as synthetically generated datasets and are summarized in Table 1.

Table 1: Benchmark and synthetic datasets

| Task | Dataset | Observations | Numbers of features | References |
|------|---------|--------------|---------------------|------------|
| **Classification** | Wine | 178 | 13 | (Forina et al., 1991). |
| | Breast Cancer Wisconsin | 569 | 30 | (Street et al., 1993) |
| | Digits | 11,797 | 64 | (Pedregosa et al., 2011) |
| | Make classification | 500 | 10 | (Pedregosa et al., 2011) |
| **Regression** | California Housing | 20,640 | 8 | (Efron et al., 2004) |
| | Diabetes | 442 | 10 | (Efron et al., 2004). |
| | Diamonds | 53,940 | 10 | (Wickham, 2008) |
| | Make regression | 500 | 10 | (Pedregosa et al., 2011) |

### B.3 LEAKAGE-SAFE CROSS-VALIDATION

We use $F$-fold cross-validation with splits $(T_f, V_f)$, $f = 1, \ldots, F$. For each selector $s$, retention level $k$, predictor $P_{t,m}$, and fold $f$:

1. fit preprocessing and selector $s$ on $\{(x_i, y_i)\}_{i \in T_f}$ to obtain $J_{s,k}^{(f)}$;

2. train $P_{t,m}$ on the projected training set $\{(\Pi_{J_{s,k}^{(f)}}(x_i), y_i)\}_{i \in T_f}$;

3. predict on the projected validation set $\{\Pi_{J_{s,k}^{(f)}}(x_i)\}_{i \in V_f}$ to obtain $\{\hat{y}_i^{(s,k,t,m,f)}\}_{i \in V_f}$.

The fold-level score is

$$S_{s,k,t,m,f} = S_t\Big(\{y_i\}_{i \in V_f}, \{\hat{y}_i^{(s,k,t,m,f)}\}_{i \in V_f}\Big), \tag{36}$$

where $S_t$ is macro-$F_1$ for classification and $R^2$ for regression. The cross-validated score is then

$$\text{Score}(s, k, t, m) = \frac{1}{F} \sum_{f=1}^{F} S_{s,k,t,m,f}. \tag{37}$$

## C KAN-BASED SELECTION INTERPRETABILITY

In this study, we visualize the internal KAN response functions $f_j(x_j)$ for the three top features on the Breast Cancer dataset Street et al. (1993), *worst concave points*, *mean concave points*, and *radius error*, as identified by **KAN-L2**, together with the corresponding malignant-class logit. This dataset is based on digitized images of fine-needle aspirates of breast masses, and the prediction task is to distinguish malignant from benign tumours from cell-nucleus morphology. Here, *concave points* quantify how many segments of a nucleus boundary are inwardly curved (concave), so *mean concave points* is the average number of such concave segments across all nuclei in an image, and *worst concave points* is the largest value observed among them. The *radius error* feature measures the variability of the nucleus radius (standard error of the mean radius) across measurements, capturing irregular or heterogeneous tumour shapes. By plotting the learned responses for these features and the resulting malignant-class logit, we can see how changes in concavity and boundary irregularity are transformed inside the network and how they ultimately influence the predicted malignancy score.

### C.1 MEAN CONCAVE POINTS INTERPRETABILITY STUDY

For *mean concave points*, the first-layer KAN response in figure 3a is roughly V-shaped. The total contribution (blue curve) is most negative around the centre of the normalized range and rises toward positive values at both low and high extremes. The base term (orange) varies smoothly and almost linearly, while the spline term (green) introduces most of the curvature, indicating that the non-linear spline component is responsible for emphasizing intermediate levels of average concavity in the hidden representation.

When this representation is propagated through the second KAN layer, the malignant-class logit in figure 4b becomes strongly skewed. Starting from low *mean concave points* (very smooth boundaries), the logit is

moderately negative (the model leans benign), increases to a clear maximum around the middle of the range (where the model assigns the highest malignancy risk), and then drops sharply for very high values of *mean concave points*. Thus, in this dataset the KAN regards lesions with intermediate average boundary concavity as most suspicious, while very smooth or extremely concave boundaries are treated as lower risk. Together, the two plots illustrate how a V-shaped internal code for this feature is mapped by the final layer into a peaked malignancy score that is highest at intermediate concavity levels.

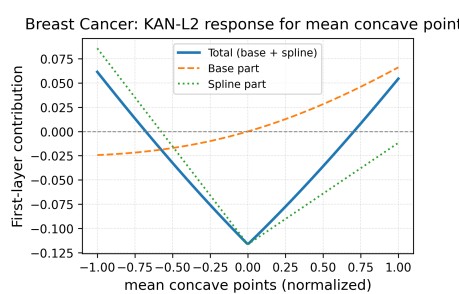 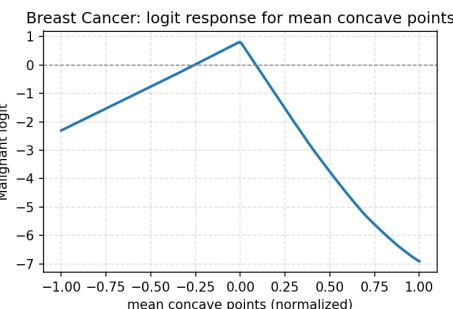

(a) KAN first-layer responses of *mean concave points*

(b) Malignant-class logit as a function of *mean concave points* (normalized), obtained by varying only this feature around a reference input.

Figure 3: Breast Cancer dataset interpretability visualizations using mean concave points.

## C.2 WORST CONCAVE POINTS INTERPRETABILITY STUDY

For *worst concave points*, the KAN layer response is monotonically decreasing (consistently with the base and spline variation), so increasing concavity systematically pushes the representation along a single direction, in line with the clinical view that strongly concave, spiculated margins are characteristic of malignant lesions. To relate this internal behaviour to the actual prediction, we also examine the malignant-class logit as a function of *worst concave points* (Figure 4). As worst concavity increases, the malignant logit rises sharply before saturating at extreme values, showing that the internal direction induced by high concavity is ultimately mapped to a higher malignancy score. Together, the first-layer responses and the logit curve indicate that KAN encodes highly concave, irregular tumour margins as a strong, monotone risk factor for malignancy, consistent with established radiological criteria.

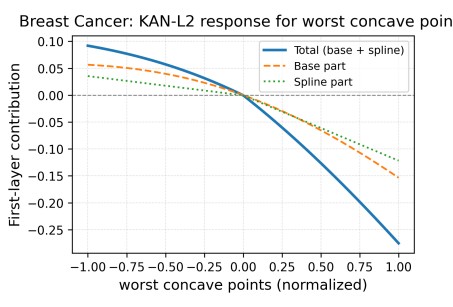 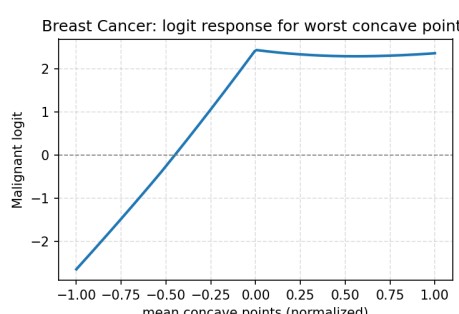

(a) KAN first-layer responses of *worst concave points*

(b) Malignant-class logit as a function of *worst concave points* (normalized), obtained by varying only this feature around a reference input.

Figure 4: Breast Cancer dataset interpretability visualizations using KAN-L2 feature choice.

## C.3 RADIUS ERROR POINTS INTERPRETABILITY STUDY

For *radius error*, the first-layer KAN response is approximately V-shaped (Figure 5a), indicating that the network uses a nonlinear code in which moderate values drive the hidden activation most strongly negative, while very small or very large errors have a weaker effect. When this representation is propagated through the second KAN layer, the resulting malignant-class logit becomes roughly monotone in *radius error* (Figure 5b): higher

radius error leads to a larger malignant logit with a mild saturation effect. This suggests that the model combines the V-shaped hidden code with other directions in feature space so that, at the prediction level, increasingly irregular tumour borders are treated as a stronger risk factor for malignancy, consistent with radiological practice.

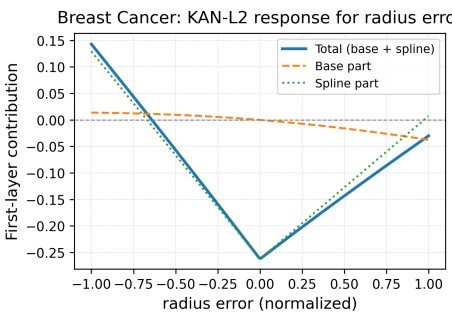
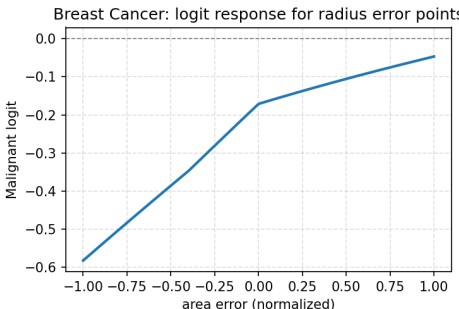

(a) KAN first-layer responses of *radius error*

(b) Malignant-class logit as a function of *radius error points* (normalized), obtained by varying only this feature around a reference input.

Figure 5: Breast Cancer dataset interpretability visualizations. (a) KAN first-layer responses for the top three features. (b) Logit sensitivity curve with respect to *worst concave points*.

## D    STABILITY AND REDUNDANCY ANALYSIS

To further assess the reliability of the proposed KAN-L2 feature selection method, we conducted two targeted diagnostic checks on representative datasets.

### D.1    STABILITY

Evaluating the reproducibility of a feature selection method is a central principle in modern feature selection, formalized in stability selection frameworks (Meinshausen & Bühlmann, 2010). To quantify how consistently each method selects the same features under small perturbations of the dataset, we measure stability across $K = 5$ cross-validation folds. For a dataset with feature matrix $X \in \mathbb{R}^{n \times d}$, let $s^{(f)} = (s_1^{(f)}, \ldots, s_d^{(f)})$ denote the feature-importance vector computed in fold $f$, and let $\alpha = 0.4$ be the retention fraction. We define the selected feature set in fold $f$ as

$$A^{(f)} = \mathrm{TopK}\Big(s^{(f)}, k\Big), \qquad k = \lfloor \alpha d \rfloor.$$

To measure agreement between folds $i$ and $j$, we use the Jaccard similarity, a standard stability index in feature selection studies (Nogueira et al., 2018),

$$J\Big(A^{(i)}, A^{(j)}\Big) = \frac{|A^{(i)} \cap A^{(j)}|}{|A^{(i)} \cup A^{(j)}|}, \quad J \in [0, 1].$$

The overall stability of a method $M$ is given by the mean Jaccard similarity across all $\binom{K}{2}$ fold pairs:

$$\mu_M = \frac{2}{K(K-1)} \sum_{1 \le i < j \le K} J\Big(A^{(i)}, A^{(j)}\Big),$$

with variability quantified by the standard deviation

$$\sigma_M = \sqrt{\frac{2}{K(K-1)} \sum_{1 \le i < j \le K} \Big(J(A^{(i)}, A^{(j)}) - \mu_M\Big)^2}.$$

Higher values of $\mu_M$ indicate greater reproducibility of the selected features, while lower $\sigma_M$ reflects reduced sensitivity to sampling noise.

Across all four variants (KAN-L1, KAN-KO, KAN-SI, and KAN-L2)(See Figure 6, the stability heatmaps show a consistent pattern: feature selection is perfectly stable on the diamonds dataset (Jaccard $\approx$ 1.00for every method), indicating that all selectors repeatedly identify essentially the same subset of features across

folds. On the more challenging digits dataset, stability decreases as expected because the feature space is larger and more redundant. Still, the KAN-based methods remain competitive with classical baselines: KAN-L1 (0.85), KAN-KO (0.49), KAN-SI (0.72), and KAN-L2 (0.62) all match or exceed the stability of Random Forest permutation importance ($\approx 0.45$), and are generally close to LASSO/L1 ($\approx 0.86$). Overall, these results show that the proposed KAN-based feature scores produce stable and reproducible feature subsets, especially when compared to existing selectors, and remain reliable even in high-dimensional settings

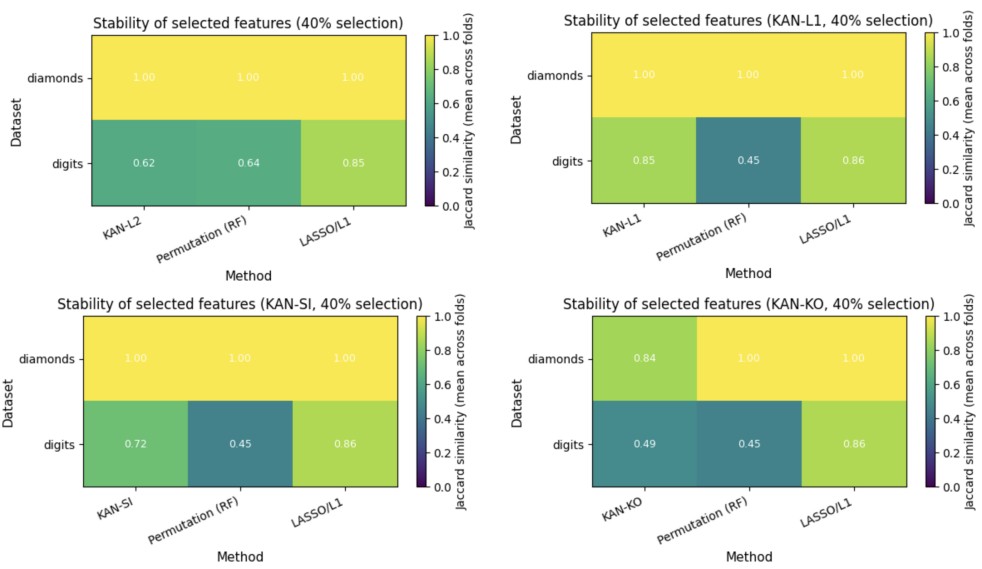

Figure 6: Stability analysis

### D.2 REDUNDANCY

To determine whether a method selects mutually correlated (and therefore potentially redundant) predictors, we compare the correlation structure of the full feature space with that of the selected subset. This follows the classical redundancy perspective underlying mRMR-style criteria (Peng et al., 2005). Let $C = \text{corr}(X) \in \mathbb{R}^{d \times d}$ be the Pearson correlation matrix of all features. The baseline redundancy level is defined as the average absolute correlation across all off-diagonal pairs:

$$\overline{|\rho|}_{\text{all}} = \frac{2}{d(d-1)} \sum_{1 \le i < j \le d} |C_{ij}|.$$

For the selected subset $S = A^{(f)}$ of size $k$, the redundancy among selected features is computed from the submatrix $C_S = C[S, S]$:

$$\overline{|\rho|}_{\text{sel}} = \frac{2}{k(k-1)} \sum_{\substack{i,j \in S \\ i < j}} |C_{ij}|.$$

A method is said to reduce redundancy when

$$\overline{|\rho|}_{\text{sel}} < \overline{|\rho|}_{\text{all}},$$

indicating that the selected predictors are less correlated than the dataset as a whole. Comparable values suggest that the selector does not inflate redundant structure, even without an explicit sparsity or decorrelation constraint. KAN variants offer complementary advantages: KAN-L2 provides the cleanest, least-redundant subsets; KAN-L1 yields sparse, efficient selections; KAN-KO balances structure and redundancy; and KAN-SI captures the strongest predictive signals even when features are correlated. Together, they offer flexible, task-adapted feature selection capabilities

Across both datasets in Figure 7, we determined the average feature correlation (all feature vs 40% selected features). The redundancy analysis shows that the KAN feature selectors behave differently depending on the selector. For diamonds, all methods start from the same baseline average correlation ($\approx 0.267$), but their selected subsets diverge significantly: KAN-L2 achieves the strongest redundancy reduction (down to 0.054), KAN-L1 and KAN-KO reduce redundancy moderately ($0.054 - 0.097$), whereas KAN-SI increases redundancy (0.328), meaning its top-ranked features are more mutually correlated. For digits, where the baseline

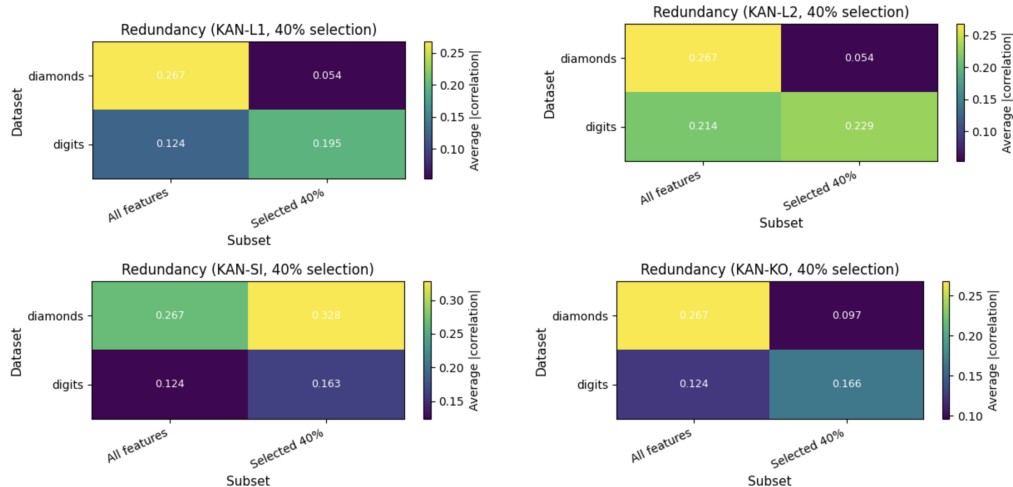

Figure 7: Redundancy analysis

redundancy is lower ($\approx 0.124$), KAN-L2 again maintains controlled redundancy (0.229), KAN-KO and KAN-SI show modest increases ($0.166 - 0.195$), while KAN-SI increases correlation the most. Overall, the results show that KAN-L2 consistently provides the least redundant subsets, KAN-L1 and KAN-KO give moderate redundancy behaviour, and KAN-SI tends to prioritise stronger, more correlated signals, which may help accuracy but does not enforce redundancy minimisation.

## E  PREDICTION PERFORMANCES PER FEATURE RETENTION LEVELS

We provided a predictive analysis in this part for different retention levels.

### E.1  AVERAGE PREDICTORS PERFORMANCES AT DIFFERENT RETENTION LEVELS

Figure equation 8 shows that Across both the Breast Cancer and Digits datasets, the heatmaps show that most selectors achieve near-saturated predictive performance once a moderate fraction of features is retained. For Breast Cancer, accuracy becomes very stable around 94-96% from roughly 30-40% feature retention onward, indicating substantial redundancy in the original predictors: using only one third of the features is almost as good as using them all. Differences between methods are most visible under aggressive pruning (10-20% retention). In this regime, LASSO/L1, Mutual Information, SVM-RFE, and the KAN-based selectors (especially KAN-L1 and KAN-SI) remain comparatively robust, whereas KAN-KO is noticeably unstable at 10% retention.

### E.2  INSIGHT AT 60% RETENTION LEVEL

In this section, we provide a comprehensive overview of the performance results of our models under different feature selection methods across all datasets considered in this study. The appendix is organized into two main subsections: the first subsection focuses on the **classification datasets**, while the second subsection presents the results for the **regression datasets**. For each dataset, we report model performance at 60% feature retention, highlighting the impact of feature selection techniques on the predictive accuracy.

#### E.2.1  CLASSIFICATION DATA SETS

This subsection presents the detailed results for all classification datasets, including breast cancer, digits, make_classification, and wine. Each table reports the macro F1 score of different predictor models using the full features data as well as reduced (60% reduction) selected feature sets obtained via KAN-based selection, LASSO, Mutual Information, Permutation importance, and SVM-RFE. The results illustrate how feature selection affects model performance and help identify the most effective selection strategies for each dataset. *Breast cancer.* (Table 2) Tree ensembles (GB, RF, XGB) remain competitive or improve with KAN-based selectors. In particular, KAN-L1 attains the top or tied-top accuracy for RF and XGB, and KAN-KO is best for GB. Logistic regression (LogReg) favors sparsity-oriented embedded/wrapper methods (SVM-RFE, LASSO), consistent with its linear inductive bias.

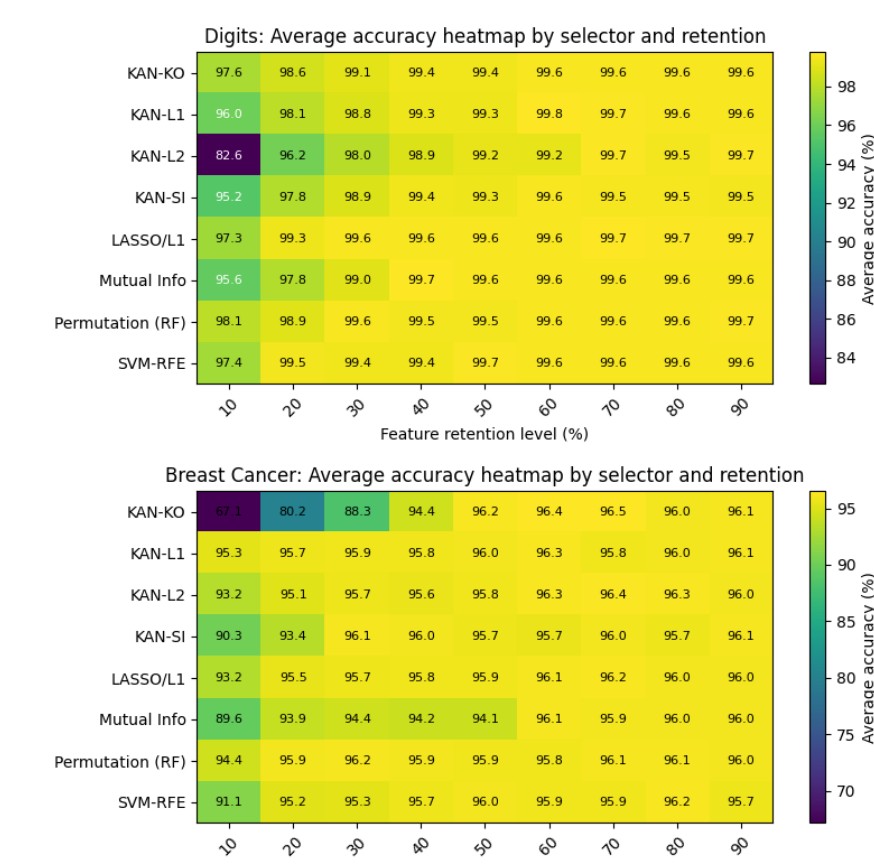

Figure 8: Average Accuracy Score (averaged across XGB, Random Forest and Gradient Boosted Trees, and Logistic Regression Classification) for 10/20/30/40/50/60/70/80/90% retained features) for the Digits and Breast cancer data sets

Table 2: 60% retention level for the *breast cancer* dataset

| Models | All Features | KAN-KO | KAN-L1 | KAN-L2 | KAN-SI | LASSO/L1 | Mutual Info | Perm. (RF) | SVM-RFE |
|--------|-------------|--------|--------|--------|--------|----------|-------------|------------|---------|
| GB | 0.9624 | **0.9698** | 0.9603 | 0.9603 | 0.9623 | 0.9603 | 0.9623 | 0.9605 | 0.9587 |
| LogReg | 0.9487 | 0.9431 | 0.9433 | 0.9489 | 0.9374 | 0.9488 | 0.9508 | 0.9376 | **0.9526** |
| RF | 0.9642 | 0.9678 | **0.9698** | 0.9661 | 0.9585 | 0.9678 | 0.9606 | 0.9584 | 0.9547 |
| XGB | 0.9716 | 0.9753 | **0.9773** | 0.9754 | 0.9698 | 0.9678 | 0.9718 | **0.9773** | 0.9681 |

*Digits.* (Table 3) Performance is near a ceiling (LogReg reaches 1.00), so differences are small. Still, KAN-L1 (and KAN-SI for GB) match or set the best scores for GB/RF/XGB, showing that KAN selectors can remove 40% of inputs without harming multiclass performance on image-like digits.

*Synthetic make_classification.* (Table 4) Moderate gains appear for RF (KAN-KO/L2) and GB (KAN-SI), suggesting that removing weak/noisy variables helps tree ensembles; LogReg again prefers sparse selectors (all strong and tied within noise).

*Wine.* (Table 5) Mixed but consistent story: KAN-L1/L2 and Permutation (RF) yield the best or tied-best for GB/XGB, while LogReg peaks with LASSO (as expected). RF is very strong overall and slightly benefits from KAN-KO.

*Takeaways.* KAN-L1/L2 are reliable selectors for nonlinear learners (GB, RF, XGB), often matching or exceeding the full-feature baseline at 60% retention. KAN-KO is competitive when measured against the end-to-end loss (notably GB and RF), validating a perturbation view of importance $\approx$. Linear LogReg benefits most from coefficient-based sparsity (LASSO, SVM-RFE). The ability to prune $\approx$ 40% of features with no loss (and

Table 3: 60% retention level for the *digits* dataset

| Models | All Features | KAN-KO | KAN-L1 | KAN-L2 | KAN-SI | LASSO/L1 | Mutual Info | Perm. (RF) | SVM-RFE |
|--------|--------------|--------|--------|--------|--------|----------|-------------|------------|---------|
| GB | 0.9869 | 0.9907 | **0.9925** | 0.9813 | **0.9925** | 0.9906 | 0.9907 | 0.9925 | 0.9907 |
| LogReg | 1.0000 | 1.0000 | 1.0000 | 1.0000 | 1.0000 | 1.0000 | 1.0000 | 0.9981 | 1.0000 |
| RF | 0.9944 | 0.9981 | **1.0000** | 0.9925 | 0.9981 | 0.9962 | 0.9962 | 0.9962 | 0.9963 |
| XGB | 0.9963 | 0.9963 | **0.9981** | 0.9926 | 0.9944 | **0.9981** | **0.9981** | 0.9963 | 0.9963 |

Table 4: 60% retention level for the *make_classification* dataset

| Models | All Features | KAN-KO | KAN-L1 | KAN-L2 | KAN-SI | LASSO/L1 | Mutual Info | Permutation (RF) | SVM-RFE |
|--------|--------------|--------|--------|--------|--------|----------|-------------|------------------|---------|
| GB | 0.8838 | 0.8879 | 0.8879 | 0.8879 | **0.8919** | 0.8617 | 0.8819 | 0.8879 | 0.8819 |
| LogReg | 0.8577 | **0.8596** | **0.8596** | **0.8596** | **0.8596** | 0.8477 | 0.8577 | **0.8596** | 0.8577 |
| RF | 0.8999 | **0.9139** | 0.9059 | **0.9139** | 0.9059 | 0.8819 | 0.8879 | 0.9098 | 0.8919 |
| XGB | 0.8959 | 0.8979 | 0.9019 | **0.9039** | 0.8999 | 0.8779 | 0.8959 | 0.8979 | 0.8959 |

Table 5: 60% retention level for the *wine* dataset

| Models | All Features | KAN-KO | KAN-L1 | KAN-L2 | KAN-SI | LASSO/L1 | Mutual Info | Permutation (RF) | SVM-RFE |
|--------|--------------|--------|--------|--------|--------|----------|-------------|------------------|---------|
| GB | 0.9452 | 0.9449 | **0.9566** | **0.9566** | 0.9458 | 0.9505 | 0.9507 | **0.9566** | 0.9450 |
| LogReg | 0.9549 | 0.9663 | 0.9611 | 0.9611 | 0.9610 | **0.9774** | 0.9663 | 0.9611 | 0.9611 |
| RF | 0.9832 | **0.9837** | 0.9676 | 0.9676 | 0.9784 | 0.9781 | 0.9729 | 0.9728 | 0.9732 |
| XGB | 0.9724 | 0.9674 | 0.9615 | 0.9615 | 0.9671 | 0.9667 | 0.9557 | 0.9670 | **0.9728** |

occasional gains) indicates substantial redundancy and supports using KAN-based selection as a practical dimensionality reduction step for tabular classification.

### E.2.2 REGRESSION DATA SETS

This subsection presents the results for all regression datasets, including California housing, diabetes, and make_regression. Each table reports the predictive performance of predictor models, measured in terms of $R^2$, under various feature selection methods using a data resulting in 60% feature retention, as well as the data having all features. The analysis emphasizes the robustness of models to feature reduction and highlights which selection methods preserve or enhance model performance. On California dataset Table 6, ensembles remain resilient at 60% retention. GB peaks with RF permutation; KAN-SI and KAN-KO track the all-features baseline closely. RF improves with KAN-L1, indicating redundancy removal helps tree splits. Ridge is fragile: all-features wins while KAN-L1/L2 over-prune complementary linear signal. XGB prefers all-features, with KAN-SI and Mutual Information nearly matching, SVM-RFE trailing slightly. Across selectors, differences remain modest.

Diabetes dataset Table 7 exemplifies small-n, noisy regression. GB favors LASSO; RF performs best with all features. Ridge is exceptionally steady near 0.41, tying All and KAN-KO, and barely changing under other selectors. XGB prefers RF permutation. KAN-L1 consistently underperforms, suggesting over-sparsification removes weak yet useful signal. Overall, cautious selection helps, but aggressive pruning hurts most predictors. Mutual Information lags across models slightly.

Diamonds dataset 8 shows a strong signal, near-ceiling performance. Ensembles (RF, GB, XGB) remain virtually unchanged; tiny gains appear with KAN-KO/L2/SI. Ridge benefits modestly from LASSO, suggesting linear redundancy. Overall, selection is largely neutral: most approaches match the all-features baseline within the third decimal. Thus, when structure and signal to noise ratio are high, compact subsets neither help nor harm meaningfully. KAN methods remain competitive On synthetic make_regression (Table. 9), signal to noise is high. Ridge sits essentially at one across most selectors, confirming linear recoverability. RF improves modestly with KAN-L2/SI; GB and XGB notch their best with SVM-RFE, though gaps are tiny. KAN variants are stable and competitive, neither overshooting nor collapsing. Overall, many selectors converge, reflecting informative features and limited redundancy. Permutation, LASSO, Mutual Information perform similarly

Table 6: 60% retention level for the *California* dataset

| Models | All Features | KAN-KO | KAN-L1 | KAN-L2 | KAN-SI | LASSO/L1 | Mutual Info | Perm. (RF) | SVM-RFE |
|--------|--------------|--------|--------|--------|--------|----------|-------------|------------|---------|
| GB | 0.7858 | 0.7825 | 0.7416 | 0.7471 | 0.7833 | 0.7565 | 0.7832 | **0.7909** | 0.7563 |
| RF | 0.8103 | 0.8117 | **0.8266** | 0.8230 | 0.8180 | 0.8257 | 0.8179 | 0.8109 | 0.8257 |
| Ridge | **0.6037** | 0.5886 | 0.3068 | 0.3933 | 0.5854 | 0.5933 | 0.5854 | 0.5946 | 0.5933 |
| XGB | **0.8431** | 0.8317 | 0.8318 | 0.8295 | 0.8409 | 0.8284 | 0.8395 | 0.8382 | 0.8261 |

Table 7: 60% retention level for the *diabetes* dataset

| Models | All Features | KAN-KO | KAN-L1 | KAN-L2 | KAN-SI | LASSO/L1 | Mutual Info | Perm. (RF) | SVM-RFE |
|--------|--------------|--------|--------|--------|--------|----------|-------------|------------|---------|
| GB | 0.4208 | 0.4157 | 0.2405 | 0.4204 | 0.4021 | **0.4326** | 0.3933 | 0.4041 | 0.4205 |
| RF | **0.4266** | 0.4032 | 0.2601 | 0.4015 | 0.4089 | 0.4132 | 0.3916 | 0.4091 | 0.4186 |
| Ridge | **0.4205** | 0.4209 | 0.2769 | 0.4027 | 0.4127 | 0.4130 | 0.4137 | 0.4127 | 0.4132 |
| XGB | 0.3591 | 0.3413 | 0.1716 | 0.3636 | 0.3432 | 0.3554 | 0.3255 | 0.3366 | **0.3682** |

*Takeaways.* KAN-SI and KAN-KO are stable selectors across regression tasks, often matching the all-feature baseline (California, make_regression). KAN-L1/L2 can benefit tree ensembles (RF, GB, XGB) by removing redundancy, but risk over-pruning in noisy, small-$n$ data (Diabetes). Ridge regression is fragile to sparsification, performing best with all features unless signal-to-noise is very high. In high-signal datasets (Diamonds, make_regression), feature selection has negligible effect, with all methods converging near the ceiling. The ability to drop $\approx 40\%$ of features without loss, and sometimes small gains, demonstrates redundancy and supports KAN-based feature selection as a practical dimensionality reduction tool for regression. For the Digits dataset, feature redundancy is even more pronounced. Many selectors already achieve 95-98% average accuracy at 10% retention, and by 30-40% retention essentially all methods converge to $\approx$ 99-99.7% accuracy, making them practically indistinguishable. The only clear weakness appears for KAN-L2 at the most extreme pruning level (10%), where performance drops to about $83\%$, but it quickly recovers once more features are kept. Overall, these results suggest that: (i) strong dimensionality reduction is possible without meaningful loss in predictive performance, especially for highly redundant datasets such as Digits; and (ii) KAN-based selectors are competitive with, and often comparable to, classical baselines such as LASSO, Mutual Information, and SVM-RFE, except for a few edge cases under very aggressive pruning.

# F  RUNTIME PROFILING OF FEATURE SELECTORS

Table 10 reports wall-clock times for computing feature scores on two representative datasets. Once a KAN has been trained, the associated selectors are relatively cheap: coefficient-based criteria (*Selector-L1*, *Selector-L2*) are as fast as or faster than LASSO, and the gradient-based score (*Selector-SI*) remains competitive with mutual information. The knock-out score (*Selector-KO*) is more expensive, as it requires repeated forward passes with ablated features, but is still considerably lighter than wrapper-style methods such as permutation importance or SVM-RFE, whose runtimes are dominated by repeated model retraining or resampling. The main overhead of our approach lies in the initial KAN training (*KAN-Train*), which is substantially slower than fitting a single linear or tree model. However, this cost is incurred once per dataset and can then be amortized across multiple selectors, retention levels, and interpretability analyses, whereas wrapper methods must pay a similar price every time a new feature subset is evaluated.

Table 8: 60% retention level for the *diamonds* dataset

| Models | All Features | KAN-KO | KAN-L1 | KAN-L2 | KAN-SI | LASSO/L1 | Mutual Info | Perm. (RF) | SVM-RFE |
|--------|--------------|--------|--------|--------|--------|----------|-------------|------------|---------|
| GB | **0.9710** | **0.9710** | 0.9708 | **0.9710** | **0.9710** | 0.9704 | 0.9707 | 0.9707 | 0.9707 |
| RF | 0.9800 | **0.9801** | 0.9792 | **0.9801** | **0.9801** | 0.9800 | 0.9800 | 0.9800 | 0.9800 |
| Ridge | 0.8805 | 0.8806 | 0.8824 | 0.8806 | 0.8806 | **0.8847** | 0.8797 | 0.8797 | 0.8797 |
| XGB | 0.9816 | **0.9817** | 0.9810 | **0.9817** | **0.9817** | 0.9810 | 0.9811 | 0.9812 | 0.9811 |

Table 9: 60% retention level for the *make_regression* dataset

| Models | All Features | KAN-KO | KAN-L1 | KAN-L2 | KAN-SI | LASSO/L1 | Mutual Info | Permutation (RF) | SVM-RFE |
|--------|--------------|--------|--------|--------|--------|----------|-------------|------------------|---------|
| GB | 0.9456 | 0.9418 | 0.9418 | 0.9501 | 0.9501 | 0.9500 | 0.9499 | 0.9500 | **0.9503** |
| RF | 0.8819 | 0.8922 | 0.8922 | **0.8976** | **0.8976** | 0.8972 | 0.8972 | 0.8972 | 0.8971 |
| Ridge | **0.99999** | 0.9885 | 0.9885 | 0.99999 | **0.99999** | **0.99999** | **0.99999** | **0.99999** | **0.99999** |
| XGB | 0.9233 | 0.9234 | 0.9234 | 0.9325 | 0.9325 | 0.9336 | 0.9341 | 0.9336 | **0.9344** |

Table 10: Selector Runtime Comparison on Diamonds and Wine Datasets

| Selector | Diamonds (sec) | Wine (sec) |
|----------|----------------|------------|
| Selector-L2 | 0.022246 | 0.001013 |
| Selector-L1 | 0.039464 | 0.001047 |
| Selector-SI | 0.125190 | 0.003804 |
| LASSO/L1 | 0.292232 | 0.001218 |
| Selector-KO | 0.426020 | 0.023628 |
| Mutual Info | 2.349357 | 0.029882 |
| Permutation (RF) | 51.709088 | 0.799606 |
| SVM-RFE | 80.940045 | 0.021224 |
| KAN-Train | 527.862040 | 2.441519 |

