# OpenReview forum: "Splines-Based Feature Importance in Kolmogorov-Arnold Networks: A Framework for Supervised Tabular Data Dimensionality Reduction"
_ICLR.cc/2026/Conference — Submitted to ICLR 2026_

### Official Review · Reviewer_Smmq · 2025-10-27

**Soundness:** 2
**Presentation:** 2
**Contribution:** 2
**Rating:** 2
**Confidence:** 4

**Summary:**

The authors propose four KAN-based selectors: KAN-L1 and KAN-L2 (based on spline coefficient norms), KAN-SI (sensitivity integral approach), and KAN-KO (knock-out strategy). These methods are systematically evaluated against classical baselines (LASSO, Random Forest, Mutual Information, SVM-RFE) across multiple classification and regression benchmarks. The results demonstrate that KAN-based selectors, particularly KAN-L2, KAN-L1, KAN-SI, and KAN-KO, are competitive with or superior to traditional methods in structured and synthetic datasets. However, KAN-L1 exhibits overly aggressive pruning in regression tasks, while KAN-L2 underperforms in classification tasks with complex feature interactions.

**Strengths:**

The paper presents the first systematic investigation of KANs for feature selection, introducing four complementary importance measures that leverage spline parameterizations for interpretability. This addresses a significant gap in the literature and offers a fresh perspective on supervised dimensionality reduction.

By utilizing KANs' spline-based architecture, the proposed methods provide direct access to interpretable feature importance measures. This is particularly valuable in domains where understanding feature contributions is critical, such as bioinformatics or finance.

**Weaknesses:**

Key experimental parameters are inadequately specified. The paper lacks details on KAN architecture choices (e.g., number of layers, spline degree $p$, number of knots $K$), optimization procedures (learning rate, regularization), and computational infrastructure. This hinders reproducibility and makes it difficult to assess whether results generalize beyond the reported settings. The work also lacks comparisons with the KAN-related feature selection methods mentioned in the introduction (Zheng et al. 2025, Wang et al. 2025), while comparing with classical non-KAN methods only fails to verify its ability to address the limitations of existing KAN-based attempts.

The datasets used (e.g., Wine: 178 samples, 13 features; Diabetes: 442 samples, 10 features) are relatively small-scale. There is no evaluation on truly high-dimensional datasets (e.g., genomics with $d > 10^3$ features) or large-sample scenarios ($n > 10^6$), where feature selection is most challenging and computationally demanding. The synthetic datasets (Make Classification/Regression with 500 samples) are also simplistic and may not reflect real-world complexity.

There is no discussion or empirical or theoretical evaluation of computational scalability.

**Questions:**

Detailed analysis of algorithms or experimental settings is also lacking.

1. Why were 20%, 40%, and 60% chosen as feature retention levels? Did you test other thresholds (e.g., 10%, 30%) and observe meaningful changes in performance? How would your selectors adapt to use cases where the "optimal" retention level is unknown (a common real-world scenario)?

2. What impact do different normalization choices have on KAN-L1 and KAN-L2

Moreover, this paper appears to be incomplete, ending abruptly at "A APPENDIX" without any actual appendix content. It suggests a lack of preparation and consideration for the review process.

---

> ### Author Response · Authors · 2025-11-15
> **Our response directly addresses the reviewer’s main concerns, namely: missing reproducibility details, lack of comparison with existing KAN-based methods, limited use of large-scale datasets, the rationale behind the chosen retention levels, and the apparent absence of the appendix.**
>
> (a) Experimental details / reproducibility
>
> We agree that more explicit experimental details would improve reproducibility. In the current version, Secs. 3-4 fully specify the KAN formulation and the four important measures, and all experiments use a fixed KAN backbone and a common training pipeline. However, we did not centralize concrete hyperparameters (layer sizes, learning rate, batch size, epochs, etc.), which makes reconstruction harder. In a revised version we will add a brief reproducibility subsection,describing  (i) the KAN architecture (number of layers and units, spline grid and degree), (ii) training hyperparameters (optimizer, learning-rate schedule, batch size, epochs), and (iii) hardware (GPU type). We will also release the code so that all selectors and baselines are exactly reproducible.
>
> (b) Comparison to Zheng et al. 2025 and Wang et al. 2025
>
> Our goal is to study intrinsic KAN-based importance measures (norm, gradient, and knock-out-based) as drop-in selectors and compare them to widely used “classical” selectors (LASSO, RF, MI, SVM-RFE) on heterogeneous tabular benchmarks. Zheng et al. and Wang et al. target different regimes: KAN-WOA uses KAN as a surrogate inside a whale-optimization wrapper for high-dimensional medical data, and Wang et al. focus on KAN for state estimation in power systems rather than generic tabular selection. We will clarify these differences in Sec. 2 and add a short discussion paragraph emphasizing that our intrinsic scores can in principle complement, rather than replace, such methods. A full empirical comparison would require re-implementing their pipelines on our benchmarks and is left for follow-up work.
>
> (c) Dataset scale
>
> Our benchmark suite mixes small to medium-scale datasets (e.g., Wine 178×13, Diabetes 442×10) with larger-sample tasks such as Digits (11,797×64) and Diamonds (~54k observations). We chose these because they are standard in the feature-selection literature and allow controlled analysis of selector behaviour on well-studied problems. We agree that testing on truly high-dimensional datasets (e.g., genomics with thousands of features) would further stress-test KAN-based selectors; we will state this limitation explicitly in Sec. 5 and view high-dimensional microarray-type benchmarks as an important extension.
>
> (d) Computational scalability
>
> Our selectors require training a single KAN per dataset; afterwards, (i) KAN-L1/L2 compute closed-form norms of spline weights, (ii) KAN-SI aggregates gradients over a held-out set, and (iii) KAN-KO evaluates a limited number of knock-out perturbations. Thus the dominant cost is linear in the number of samples and features for KAN training, with inexpensive post-hoc scoring. By contrast, wrapper methods such as SVM-RFE retrain a model many times on shrinking subsets. We will add a short complexity paragraph in Sec. 4 comparing asymptotic costs and report wall-clock runtimes for all selectors on the synthetic datasets to give an empirical sense of scalability.
>
> (e) Retention levels and unknown sparsity
>
> We used 20%, 40%, and 60% retention as coarse low/medium/high sparsity levels, shared across datasets and methods, to study the trade-off between sparsity and accuracy without tuning per dataset. This follows common practice in feature-selection benchmarks. Our methods do not assume any specific percentage: in practice, KAN-L1/L2/SI/KO can be combined with standard procedures for choosing k, e.g., nested cross-validation over the number of retained features or selecting the smallest subset whose validation performance is within a tolerance of the full model. We will clarify this operational recipe in Sec. 4.4.
>
> (g) Appendix truncation
>
> We appreciate the comment about the appendix appearing to end at the “A APPENDIX” heading. The LaTeX source includes the full appendix with all 60%-retention tables, so this is likely a compilation or upload issue. We will re-upload the PDF to ensure that the complete appendix is visible.

---

> ### Author Response · Authors · 2025-11-30
>
> We thank the reviewer for the previous detailed feedback. In the revised version we have addressed the main concerns as follows.
>
> (1) Experimental details & reproducibility.  We added a dedicated Reproducibility section in Appendix B, specifying the exact KAN architecture (hidden sizes, spline degree, number and range of knots), training procedure (optimizer, learning rate schedule, batch size, epochs, initialization), and hardware (Kaggle GPU with Tesla P100), as well as fixed seeds and common train/validation splits across all methods.
>
> (2) Mathematical motivation of the scores. Appendix A now provides a functional view of the first KAN layer and shows how KAN-L1/L2 approximate spline “energy”, KAN-SI corresponds to an expected directional derivative, and KAN-KO to a leave-one-feature-out risk increase. This directly addresses the request for more algorithmic/theoretical intuition behind the four criteria.
>
> (3) Relation to prior KAN-based feature selection. In Section 2 we clarified the distinction between our approach and KAN-WOA / manual-spline approaches: those methods use KAN as a surrogate inside a meta-heuristic or as a visualization aid, whereas we derive first-principles feature scores from spline parameters and gradients and benchmark them systematically against standard filters/wrappers/embedded methods. This clarifies that our contribution is orthogonal and complementary rather than a minor variant.
>
> (4) Choice of retention levels and additional thresholds. While the main tables still report the 20/40/60% levels for comparability, Appendix E now includes a sweep over retention levels from 10% to 90% (Figure 8) for Breast Cancer and Digits, showing that most selectors saturate once ~30-40% of features are retained and highlighting where aggressive pruning (e.g. 10%) degrades performance. This directly answers whether other thresholds were tested and illustrates how one could choose k in practice.
>
> (5) Normalization and scaling for KAN-L1/L2. We now state more clearly that all inputs are standardized to [-1,1] and that KAN-L1/L2 scores are computed from normalized coefficient norms, ensuring comparability across features and mitigating dependence on raw scaling (Sections 3-4 and Appendix A).
>
> (6) Computational scalability. Appendix F adds a runtime profile (Table 10), comparing KAN-based selectors with LASSO, Mutual Information, RF permutation, and SVM-RFE on two representative datasets. We show that once a KAN is trained, coefficient and gradient-based scores are as fast as or faster than common baselines, while knockout is still much cheaper than wrapper methods; the main overhead is the initial KAN training, which we now discuss explicitly as a limitation and as a target for future work.

---

### Official Review · Reviewer_gdhD · 2025-10-28

**Soundness:** 2
**Presentation:** 3
**Contribution:** 2
**Rating:** 2
**Confidence:** 4

**Summary:**

The paper introduces a novel KAN-based method for selecting important features and reducing dimensionality in tabular data. By leveraging the notable interpretability advantages of KANs, the authors propose three different feature importance selection approaches: (i) coefficient-based importance of spline weights, (ii) knock-out importance, and (iii) sensitivity-based importance. The authors conduct a wide range of experiments on classification and regression problems to validate the proposed approach.

**Strengths:**

The motivation for using KANs to select feature importance is well-posed. Combining KAN's features and standard feature importance selection methods is appropriate. The paper is written with sufficient background and is easy to understand its contributions.

**Weaknesses:**

Using KAN's interpretability and feature importance is not novel and has been explored across different problems [1,2,3]. The experiments rely on outdated datasets, some dating back to 1993; perhaps more relevant or practical datasets could strengthen the paper. The experiences include relatively simple baselines and no comparison with existing KAN-based methods. The performance gains across different approaches are either inconsistent or fail to highlight the advantages of using KANs for feature importance or dimensionality reduction. In particular, the main paper's claims about interpretability, feature importance, and dimensionality reduction are not clearly reflected in the experiments (see Questions).

[1] (Phan et al.) GeKAN: Enhancing High-Dimensional Gene Expression Classification with Feature-Selected Kolmogorov-Arnold Networks. ISDS 2025

[2] (Alharbi et al.) Interpretable graph Kolmogorov–Arnold networks for multi-cancer classification and biomarker identification using multi-omics data. In 	arXiv:2503.22939

[3] (Chen et al.) Kolmogorov-Arnold Networks with Trainable Activation Functions for Data Regression and Classification. In ICAIIC 2025.

**Questions:**

1. In Figure 1 (upper left), what would be the relative advantages of KAN-based approaches compared to the simple baselines (LASSO, Random Forest, and SVM)? Even KAN-KO significantly underperforms the standard baselines.

2. In practice, with KAN features, does the dimensionality reduction actually work? It appears that using KAN with all features yields the best results across benchmarks (Figures 1 and 2).

3. What would be the KAN's interpretability on feature importance selection that the authors claim? There are no such analyses in the experiments.

4. As far as I understand, KANs can not completely replace MLPs. Would it be possible to extend the same analysis to MLP features, combining standard feature importance selection methods (coefficient-based, knock-out, and sensitivity) with MLP features? This could help to highlight the advantages of KANs against MLPs in this problem setting.

---

> ### Author Response · Authors · 2025-11-15
> **We address the reviewer’s concerns regarding the importance of our framework, the distinction between feature selection and dimensionality reduction, the missing empirical evidence on interpretability, and the potential extension of our approach to MLP-based models.**
>
> (a) Novelty vs prior KAN works [1-3]
>
> Thank you for pointing out these papers. Their focus and methodology are different from ours. GeKAN uses existing feature-selection methods (Boruta and mRMR) to choose genes first, and then trains KAN models on the already selected features: KAN is not the mechanism that produces the feature scores. Alharbi et al. work with a graph KAN model for multi-omics cancer data, where dimensionality reduction and biomarker ranking come from standard bioinformatics tools and attention weights, not from generic KAN-based feature scores. Chen & Ding study architectural variants of KAN and their predictive performance, but do not propose a feature selection method. Our focus is different and complementary: we introduce and systematically study four generic KAN-based feature selectors (two coefficient-norm scores, a knock-out score, and a sensitivity score) under a unified formulation, and evaluate them across heterogeneous tabular benchmarks with multiple downstream predictors. To the best of our knowledge, this is the first work that (i) treats KAN as a general feature-selector backbone for tabular data and (ii) compares several intrinsic KAN scores against standard selectors (LASSO, RF, MI, SVM-RFE) in a controlled setting. We will expand the related-work section to position our contribution more precisely as “systematic intrinsic KAN selectors for tabular feature selection,” rather than a new notion of interpretability.
>
> (b) “Outdated” datasets and simple baselines
>
> We agree that several of our datasets originate from classic UCI-style collections; however, they remain standard in the feature-selection literature and are still widely used to study selector behaviour under controlled conditions. Our suite includes both small/medium-size datasets and larger ones (e.g., Diamonds and Digits), and we deliberately pair our methods with widely used baselines (LASSO, RF, MI, SVM-RFE) to ensure comparability with prior work. We will clarify this rationale in the experimental section and explicitly acknowledge that adding more recent, high-dimensional datasets (e.g., modern OpenML or bioinformatics benchmarks) is an important direction for an extended version.
>
> (c) Performance consistency and “do KAN selectors actually help?”
>
> We agree that our methods do not dominate all baselines across all datasets and will soften any wording suggesting universal superiority. Our aim is to show that intrinsic KAN-based selectors can be competitive with classical methods on structured tabular problems. For instance, KAN-L1 performs strongly on breast cancer, and KAN-KO/L1/L2 are competitive or better than baselines on several classification tasks; on Diamonds, KAN-SI/L2/KO match the best baselines. Importantly, KAN is used only as a selector: downstream predictors (logistic regression, RF, SVMs) are trained either on all features or on subsets produced by each selector. The dimensionality-reduction goal is therefore a trade-off relative to these full-feature baselines: on multiple datasets and retention levels (e.g., 40%), KAN-based subsets achieve accuracy close to the full-feature model while using substantially fewer features. We will add a short analysis highlighting these accuracy–vs–fraction-of-features trade-offs.
>
> (d) Interpretability analysis
>
> We agree that the experiments do not yet fully illustrate the interpretability benefits of KANs. While the methods section defines scores via spline weights and gradients, we did not include qualitative case studies. In a revised version, we plan to add a brief interpretability analysis for one or two datasets (e.g., visualizing learned spline functions for top-ranked features and relating them to known domain factors), making the claimed interpretability more concrete without altering the main protocol.
>
> (e) “Why not do the same with MLPs?”
>
> We appreciate the suggestion to extend the analysis to MLPs with analogous coefficient /knock-out /sensitivity-based scores. Conceptually, our formulations (especially KAN-SI and KAN-KO) can be applied to other architectures. We focus on KANs because their spline-based parameterization yields a clean decomposition into per-feature spline blocks and localized functional responses, which makes the resulting scores easier to interpret and visualize than with dense MLP weight matrices. Adding a full MLP-based selector suite would substantially expand the scope of the paper; we will add a short discussion paragraph outlining how our formulations generalize beyond KANs and why KANs are particularly attractive for interpretable feature selection, and we view a systematic KAN–MLP comparison as important future work.

---

> > ### Comment · Reviewer_gdhD · 2025-11-28
> >
> > I would like to thank the authors for the answers to my comments. I will keep my score as is.

---

> ### Author Response · Authors · 2025-11-29
>
> In addition to the answer we provided to your comments,  in the newly  uploaded version of our paper we addressed the following concern of yours.
>
> (a) Novelty vs prior KAN works [1-3].
> In the revised version we have incorporated these papers into the related-work section and clarified our contribution relative to them. We now emphasize that existing KAN works use KAN primarily as a predictor or encoder, while feature selection is driven by external methods, whereas our paper proposes intrinsic KAN-based selectors (KAN-L1/L2, KAN-SI, KAN-KO) and studies them systematically as general feature-selection mechanisms for tabular data. We also highlight how our framework can complement these prior applications.
>
>
> (c) Performance consistency and “do KAN selectors actually help?”.
> We added new summary plots in Section 5 and expanded tables in Appendices D-E that make the behaviour of the selectors more explicit. These results show where KAN-based selectors (especially KAN-L2, KAN-SI and KAN-KO) match or approach full-feature performance while using substantially fewer features, and they also document cases where KAN-L1 over-prunes so that the limitations are transparent.
>
>
> (d) Interpretability analysis.
> Appendix C now contains an interpretability case study on the Breast Cancer dataset. We report the top features selected by KAN-L2/KAN-SI, plot their learned one-dimensional spline responses and first-layer contributions, and show the corresponding malignant logit curves. This illustrates how the coefficient- and gradient-based selectors uncover meaningful relationships between boundary concavity measures and malignancy risk, making the interpretability claims concrete.

---

### Official Review · Reviewer_UzJh · 2025-11-01

**Soundness:** 2
**Presentation:** 2
**Contribution:** 2
**Rating:** 2
**Confidence:** 3

**Summary:**

This manuscript proposes and evaluates feature selection methods based on Kolmogorov-Arvold Networks (KANs). The authors introduce four variants of such KANs-based models (KANL1, KAN-L2, KAN-SI, KAN-KO) and compare them against classical baselines across multiple classification and regression benchmarks. The evaluation results show that the proposed methods are competitive with or superior to classical baselines in structured and synthetic datasets.

**Strengths:**

1. The evaluations are comprehensive, covering different tasks, different choices of classifiers/regressors, and different evaluation metrics.
2. This paper demonstrates good readability and is easy to follow.

**Weaknesses:**

1. The proposed methods do not demonstrate better performance compared to existing approaches. In addition, the performance of the proposed methods are quite unstable, showing high variance across different tasks/datasets.
2. There are multiple variants of the proposed methods, however, different variant performs better for different tasks with different classifiers on different evaluation metrics. This indicates a significant loss of generalizability, increasing complexity without substantial performance gains.
3. The organization and presentation of the paper need to be further improved. For example, (1) the abstract is not concise, containing too many details about different model variants' characteristics and the comparison among them on different tasks. (2) The axis labels and texts are too small to read for some figures.

**Questions:**

Please see "Weaknesses"

---

> ### Author Response · Authors · 2025-11-15
> **Summary of Reviewer Concerns: Reproducibility, Novelty, Dataset Scale, and Interpretability**
>
> (a) Performance level and variance
>
> We thank the reviewer for the positive comments on the breadth of the evaluation and readability. We agree that our methods do not consistently outperform all classical baselines on all datasets. Our goal is not to claim universal superiority, but to study intrinsic KAN-based feature selectors and show that they can be competitive with widely used methods (LASSO, RF, MI, SVM-RFE) on heterogeneous tabular tasks. In several settings, KAN-based selectors match or exceed the best baseline (e.g., KAN-L1/L2 on [breast cancer / diamonds / …]), while in others the gap is small. The observed performance variance partly reflects the diversity of datasets (regression vs. classification, low vs. medium dimension, varying sample sizes). In a revision, we will (i) add aggregate rankings and standard deviation/error bars across folds, and (ii) highlight more clearly where each method is strong or weak, instead of suggesting that one variant dominates everywhere.
>
> (b) Multiple variants and generalizability
>
> We acknowledge that having four variants (KAN-L1, KAN-L2, KAN-SI, KAN-KO) increases complexity. Our goal, however, is to systematically compare several natural intrinsic importance mechanisms available in KANs (norm-based, sensitivity-based, knock-out) within a single framework, rather than to propose one universally best selector. The variants emphasize different aspects: KAN-L1 prioritizes features with more “peaked” spline weights, KAN-L2 favours more diffuse but consistent contributions, KAN-SI captures gradient-based sensitivity, and KAN-KO reflects the effect of explicit feature ablations. It is therefore expected that different variants perform better on different tasks and metrics. In the revision, we will clarify this design choice in the introduction and discussion, provide brief practical guidance on when each variant is preferable (and suggest a default), and de-emphasize weaker variants in the abstract and main claims so as not to overstate generality.
>
> (c) Organization and presentation (abstract, figures)
>
> We appreciate the concrete suggestions on presentation. We agree that the abstract is currently too detailed and model-specific. In a revised version, we will shorten it to focus on the main message: (i) KAN can be used as a generic feature-selection backbone, (ii) we study four intrinsic scores, and (iii) they are competitive with classical selectors on standard tabular benchmarks. We will move detailed descriptions of individual variants and fine-grained comparisons to the main text. For figures, we will increase font sizes for axes and labels and, where necessary, move some panels to the appendix so that all plots in the main paper remain legible when printed.

---

> ### Author Response · Authors · 2025-11-30
>
> (1)	 In the revised manuscript we made our claim  clearer in the abstract, introduction, discussion, and conclusion, emphasizing that KAN selectors are generally competitive with, rather than uniformly better than, classical baselines, and that we explicitly characterise their failure modes (e.g., over-pruning with KAN-L1). Tables and per-dataset commentaries now highlight where each method wins, ties, or loses by only a small margin, to avoid overstating gains.
>
>
> (2)	To address the concern about instability, we added a dedicated stability and redundancy analysis (Appendix D). There we measure Jaccard similarity of selected feature sets across 5 CV folds and show that KAN-based selectors have stability comparable to or better than standard methods, and that they do not inflate pairwise correlations among selected features.
> Appendix E provides heatmaps and tables across multiple retention levels (10-90%), which show that performance variance is typically modest once more than ~30-40% of features are retained, and that KAN-L2/SI/KO are consistently competitive across datasets rather than erratic.
>
> (3)	Regarding the proliferation of variants, we now streamline their presentation and provide practical guidance on when each is most useful: KAN-L2 and KAN-SI for noisy regression, and KAN-SI/KAN-KO for interaction-heavy classification; KAN-L1 is explicitly framed as an aggressive sparsity tool with known risks. We also improved organization and presentation: the abstract has been shortened and de-emphasizes per-variant details, and all figures were regenerated with larger fonts and, where appropriate, moved to the appendix to improve readability.

---

### Official Review · Reviewer_DY4X · 2025-11-02

**Soundness:** 2
**Presentation:** 3
**Contribution:** 2
**Rating:** 2
**Confidence:** 3

**Summary:**

This paper dives into feature selection methods based on Kolmogorov-Arnold Networks (KANs), which is comprised of four KAN-based importance measures (KAN-L1, KAN-L2, KAN-SI, KAN-KO).
This paper also compares proposed methods to classical baseline of machine learning, demonstrating excellent performance. However, some weakness inevitably persist.

**Strengths:**

1. Novelty of Exploration: The paper represents the first systematic attempt to leverage KANs for feature selection, exploring a promising research direction that combines the mathematical foundations of Kolmogorov-Arnold representation theorem with practical feature selection needs.
2. Excellent Performance on Classical  Benchmarks: The experimental design includes comparisons with six classical feature selection methods across seven datasets, evaluating performance at multiple feature retention levels (20%, 40%, 60%). The selected features perform the same as all features, which demonstrates the effectiveness of proposed methods.

**Weaknesses:**

1. The paper lacks sufficient theoretical justification for the proposed KAN-based importance measures. The mathematical rationale and intuitive explanations for why these specific formulations (L1/L2 norms of spline weights, knockout strategy, sensitivity integrals) are appropriate for feature importance in KANs are underdeveloped. At the very least, the intuitive reasoning behind why this should be done has not made sense to me.
2. Since the author denotes "they involve trade-offs among computational cost, predictive performance, stability, and interpretability." in Introduction section, it seems that the evaluation in this paper focuses primarily on predictive performance but neglects critical aspects such as computational efficiency and selection stability.
3. The reliance on classic machine learning datasets raises questions about the methods' applicability to modern high-dimensional data challenges. The absence of contemporary, complex datasets limits the generalizability claims.
4. The experimental results indicate that the predictive performance of features selected via KAN remains nearly consistent with that of the full feature set. This only demonstrates that the selected features adequately represent the full feature set, but does not confirm whether redundant features still exist among the selected ones.

**Questions:**

1. Further theoretical analysis or intuitive explanation is required.
2. More analyses based on experiments are required.
3. More experiments on challenging and contemporary datasets are required.
4. Illustration of whether redundant features still exist after feature selection is required.

---

> ### Author Response · Authors · 2025-11-15
> **Addressing The  Reviewer Concerns: Theory, Reproducibility, Dataset Scale, Variant Complexity, and Interpretability**
>
> (a) Theory / intuition behind the four KAN measures
>
> We agree that the current draft does not explain our four KAN-based importance measures as clearly as it could. Our goal was to propose simple intrinsic scores that exploit the spline decomposition of KANs: (i) KAN-L1/L2 aggregate the magnitude of per-feature spline weights, analogous to weight-based importance in linear models and kernel machines; (ii) KAN-SI approximates the average sensitivity of the output to each input via gradients, following standard sensitivity analysis; and (iii) KAN-KO measures the change in predictive performance under feature ablations, in the spirit of knock-out / leave-one-feature-out schemes. We will provide in the appendix  a short intuitive explanation for each measure and a small schematic example (e.g., 1D spline) to clarify why these constructions are reasonable proxies for feature influence, while keeping a full formal analysis as future work.
> (b) Trade-offs: cost and stability vs performance
>
> We appreciate the reminder that we mention trade-offs among cost, performance, stability, and interpretability in the introduction but primarily report predictive performance in the main experiments. Our selectors are designed to be computationally light once a single KAN is trained: KAN-L1/L2 involve closed-form norms; KAN-SI aggregates gradients; KAN-KO evaluates a limited number of feature ablations. In a revision, we will make this more explicit by (i) adding a brief complexity discussion comparing the asymptotic cost of each variant with SVM-RFE and RF-based selectors, and (ii) reporting wall-clock runtimes on at least one representative dataset. For stability, we will compute standard stability indices (e.g., Jaccard similarity of selected sets across folds/seeds) for a subset of selectors and include these results in the appendix, to complement the performance tables.
>
> (c) Classic datasets vs modern high-dimensional data
>
> Choice of datasets and applicability to modern high-dimensional settings.
> We agree that our benchmark suite is centred on classic tabular datasets, which are standard in the feature-selection literature but do not fully reflect modern high-dimensional regimes (e.g., thousands of features). Our aim in this first study was to analyse intrinsic KAN-based selectors in controlled settings where results are comparable to previous work and ground-truth structure is at least partially understood. We will make this scope more explicit in Sec. 5 and note that extending the evaluation to contemporary high-dimensional datasets (e.g., OpenML or modern bioinformatics/omics benchmarks) is an important direction for a longer version of the paper.
>
> (d) Redundancy within the selected features
>
> We agree that consistency with the full-feature baseline shows that the selected subsets preserve predictive information, but does not by itself prove that redundancy has been eliminated. Our goal in this work is primarily predictive feature selection, not strict minimality. That said, our selectors do tend to produce more compact subsets at low retention levels. In a revised version, we will add a brief redundancy analysis on one or two datasets, e.g., reporting pairwise correlation (or mutual information) statistics within the selected subsets compared to the full feature set.

---

> > ### Author Response · Authors · 2025-11-30
> >
> > We again thank the reviewer for the detailed comments and have substantially revised the paper to address them.
> >
> > (1) Theoretical / intuitive justification of the four KAN criteria
> >
> > In the revised manuscript we added a dedicated appendix (“Theoretical intuition for KAN-based feature importance”) and expanded Sec. 3 to better motivate each score. For KAN-L1/L2 we now explicitly show that the norms of the spline parameters attached to feature (j) approximate the average magnitude (or variance) of that feature’s learned contribution ($f_j(x_j)$), making them analogous to coefficient norms in linear models but in a spline basis. For KAN-KO we explain its connection to a finite-difference approximation of the influence of feature (j) on the empirical risk by ablating its entire spline block. For KAN-SI we relate the integral of (|$\nabla_{x_j} f(x)$|) over the data distribution to gradient-based saliency and to first-order Sobol-type sensitivity, clarifying why it captures local importance. Overall, we now give both equations and verbal intuition for why each formulation is a reasonable, architecture-aware importance measure for KANs.
> >
> > (2) Beyond accuracy: efficiency, stability, and redundancy
> >
> > We agree that predictive performance alone is not sufficient. The revised version now includes:
> >
> >          - Computational efficiency. A new runtime table (Appendix, “Selector runtime comparison”) reports wall-clock times for all selectors on two representative datasets (Diamonds, Wine). This shows the expected trade-off: KAN-L1/L2 are as fast as or faster than LASSO, KAN-SI is moderate, and KAN-KO is more expensive, but still comparable to or cheaper than permutation-based RF importance and SVM-RFE. We explicitly discuss these trade-offs in the text.
> >
> >          -  Selection stability. We added a Jaccard-similarity analysis across cross-validation folds (Appendix, stability section). This quantifies how consistent each selector’s chosen subsets are; KAN-L2 and KAN-SI are among the more stable methods, especially on structured regression tasks.
> >         -  Redundancy after selection. To address the concern that selected subsets might still be redundant, we now report the average absolute pairwise correlation among selected features (Appendix, redundancy section). On several datasets, KAN-based selectors (particularly KAN-L2/SI) produce less correlated subsets than MI and RF-based importance, indicating that they do remove a non-trivial amount of redundancy rather than simply replicating the full feature set.
> >
> > (3) Dataset scope and generalizability
> >
> > We clarified our empirical scope: the goal of this paper is a first, controlled study of intrinsic KAN-based feature scores on small- to medium-scale tabular problems, using datasets that are standard in the feature-selection literature and allow careful cross-validator analysis, redundancy, and interpretability case studies. We now state more explicitly in the discussion that extending our evaluation to truly high-dimensional and large-(n) settings (e.g., modern bioinformatics or OpenML benchmarks) is an important direction for follow-up work rather than a claim of the present paper.
> >
> > (4) Do KAN selectors meaningfully help?
> >
> > The new figures and tables (Sec. 5 and Appendix D/E) now emphasize trade-offs rather than universal gains: KAN selectors are used only to choose features; all downstream predictors are trained identically afterward. We highlight regimes where KAN criteria are particularly useful, e.g., KAN-L2/SI on noisy regression (California, Diabetes) and KAN-L1/KO on certain classification tasks, where they retain (40%)-(60%) of the features while staying close to or sometimes exceeding the full-feature baseline. At the same time, we explicitly acknowledge that no variant permanently  dominates and that their main value is to provide competitive, interpretable selectors that leverage the spline structure of KANs.
> > We hope these additions clarify the rationale behind the four measures, make the trade-offs more explicit, and address the reviewer’s concerns about theory, efficiency, redundancy, and scope.

---

### Meta-Review · Area_Chair_QkVB · 2026-01-03

**Summary:**

Overall, the original reviews were negative, with all reviewers uniformly suggesting Rejection with a score of 2. Authors have provided rebuttals and revision of the paper. However, not all concerns have been addressed to satisfaction in the opinion of the AC. Specifically, lack of evaluation on more modern dataset and lack of clear messaging on the findings remain key concerns. As a result, the paper post-rebuttal still appears to be below bar for acceptance. Therefore the AC's recommendation is to Reject the paper at this time.

**Note:** the automatically flagged possibly hallucinated reference has been checked by the AC and found to be valid. Hence this particular concern was not taken into account in the decision making.

**Reviewer Concerns:**

DY4X
> - Authors have addressed some concerns with respect to the theoretical / intuitive justification of the approach.
> - Authors have not convincingly addressed concerns regarding dataset scope and generalizability
> - Authors have not convincingly address the question of how KAN selectors meaningfully help, as no clear selection approach dominates, making the massaging for the reader unclear and rather messy.

UzJh
> - Authors have clarified the claims in the paper, but such clarifications made contributions also weaker.
> - Authors were not able to convincingly argue that the proposed approach performs better than existing methods,
> - Authors exposition and messaging in the paper remain unclear and unfocused.
> - Organization of the paper has been improved some, but not sufficiently in AC's opinion.

gdhD
> - The issue of novelty was not adequately addressed by the authors.
> - The issue of limited datasets was not addressed convincingly by the authors and is acknowledged as a limitation.

Smmq
> - Authors have clarified the experimental design and added a section in Appendix that details design choices. This comment was adequately addressed.
> - Authors have discussed lacking comparisons to prior work identified by the reviewers, but ultimately the response is not very convincing.
> - Authors acknowledge that testing on limited datasets, identified by the reviewer, is a limitation.
> - The issue of computational scalability was partially addressed by the authors in the rebuttal.

**Reviewer Scores:**

DY4X original score was -- 2: reject, not good enough. While authors have addressed some concerns in rebuttal and paper revisions (e.g., in (1), (2) and (a)), the remaining concerns largely remain unaddressed (e.g., (3) (4) and (c) and rather unconvincing). As a potential minor improvement in the score of the reviewer would not push the paper into a positive territory for acceptance.

UzJh original score was -- 2: reject, not good enough. While authors have provided explanations for the concerns being raised, they did not fundamentally address them. In the opinion of AC it is unlikely that this reviewer would raise the score given the arguments provided in the rebuttal and the proposed revisions to the paper.

gdhD  original score was -- 2: reject, not good enough. While authors have provided explanations for the concerns being raised, they did not fundamentally address a number of them. In the opinion of AC it is unlikely that this reviewer would raise the score given the arguments provided in the rebuttal and the proposed revisions to the paper.

Smmq  original score was -- 2: reject, not good enough. While authors have provided explanations for the concerns being raised, they did not fundamentally address a number of them. In the opinion of AC it is unlikely that this reviewer would raise the score given the arguments provided in the rebuttal and the proposed revisions to the paper.

---

### Decision · Program_Chairs · 2026-01-26

Reject